# Generalization in Deep RL for TSP Problems via Equivariance and Local Search

## Abstract

Deep reinforcement learning (RL) has proved to be a competitive heuristic for solving small-sized instances of traveling salesman problems (TSP), but its performance on larger-sized instances is insufficient. Since training on large instances is impractical, we design a novel deep RL approach with a focus on generalizability. Our proposition consisting of a simple deep learning architecture that learns with novel RL training techniques, exploits two main ideas. First, we exploit equivariance to facilitate training. Second, we interleave efficient local search heuristics with the usual RL training to smooth the value landscape. In order to validate the whole approach, we empirically evaluate our proposition on random and realistic TSP problems against relevant state-of-the-art deep RL methods. Moreover, we present an ablation study to understand the contribution of each of its components.

## 1 Introduction

The traveling salesman problem (TSP) has numerous applications from supply chain management (Snyder & Shen, 2019) to electronic design automation (Gerez, 1999) or bioinformatics (Jones & Pevzner, 2004). As an NP-hard problem, solving large-sized problem instances is generally intractable. Deep reinforcement learning (RL)-based heuristic solvers have been demonstrated (Bello et al., 2016; Dai et al., 2017; Kool et al., 2019; Ma et al., 2019) as being able to provide competitive solutions while being fast, which is crucial in many domains such as logistics for real-time operations. However, this ability has only been demonstrated on small-sized problems where RL-based solvers are usually trained and tested on small instances. When evaluated instead on larger instances, such solvers perform poorly. Since the computational cost of training on large instances is prohibitive, increasing the size of the training instances is not a practical option for obtaining efficient solvers for larger instances. To overcome this limitation, this paper investigates techniques to increase the generalization capability of deep RL solvers, which would allow to train on small instances and then solve larger ones.

Generalization in deep RL (François-Lavet et al., 2018) can be improved by adjusting the following three components: input/representation space, objective function, and learning algorithm. We propose to achieve this by exploiting equivariance, local search, and stochastic curriculum learning, which we explain next. Although our approach is demonstrated on TSP problems, we believe it can be naturally extended to other combinatorial optimization problems, but also to more classic RL domains. We leave this for future work.

Problems with spatial information, such as TSP problems, enjoy many spatial symmetries which can be exploited for RL training and generalization. An RL solver is invariant with respect to some symmetry, if its outputs remain unchanged for symmetric inputs. For instance, translating all the city positions leave optimal solutions intact. More generally, an RL solver is equivariant, if its outputs are also transformed with a corresponding symmetry for symmetric inputs. For instance, if cities are permuted, a corresponding permutation is required on the outputs of the RL solver. Invariance and equivariance of the solver with respect to some symmetries allows a smaller input space to be considered during training and more abstract representation insensitive to those symmetries to be learned while the trained solver still covers the whole input space.

Previous work has considered using local search as a simple additional step to improve the feasible solution output by a RL-based solver. In contrast to most previous work, we further employ local

search as a tool to smooth the objective function optimized during RL training. To that aim, we interleave RL training with local search using the improved solution provided by the latter to train the RL solver via a modified policy gradient. Moreover, we propose a novel simple baseline called the policy rollout baseline to reduce the variance of its estimation.

Curriculum learning (Soviany et al., 2021) can accelerate training, but also improve generalization (Weinshall et al., 2018). By stochastically adjusting the difficulty of the training instances, as described by instance sizes, the RL-based solver can learn more easily and faster. Since the solver sees various sizes of instances, this may prevent it to overfit to one particular instance size.

The architecture of our model includes a graph neural network (GNN), a multi-layer perceptron (MLP), and an attention mechanism. Due to all the used techniques, we name our model as eMAGIC (e for equivariance, M for MLP, A for Attention , G for Graph neural network, I for Interleaved local search, and C for Stochastic Curriculum learning). We demonstrate that it can be trained on small random instances (up to 50 cities) and perform competitively on larger random or realistic instances (up to 1000 cities). The only learning methods that can perform better than ours require learning on the instance to be solved (see Section 2 for a discussion).

## 2 RELATED WORK

Research on exploiting deep learning (Vinyals et al., 2015; Li et al., 2018; Joshi et al., 2019a; Prates et al., 2019; Xing & Tu, 2020) or RL (Bello et al., 2016; Dai et al., 2017; Kool et al., 2019; Deudon et al., 2018; Ma et al., 2019; Da Costa et al., 2020; Zheng et al., 2021; Kwon et al., 2021; Wu et al., 2021b) to design heuristics for solving TSP problems has become very active. The current achievements demonstrate the potential of machine-learning-based approaches, but also reveal their usual limitation on solving larger-size instances, which has prompted much research on generalization (Joshi et al., 2019b; Fu et al., 2021; Wu et al., 2021b). Some experimental work (Joshi et al., 2019b) suggests that deep RL may provide more generalizable solvers than supervised learning, which is a further motivation for our own work. Another advantage of deep RL is that it does not require optimal solvers to train. We discuss next the related work that exploits equivariance or local search like our method. To the best of our knowledge, no other work uses **stochastic** curriculum learning for designing an RL-based solver for TSP, although Lisicki et al. (2020) evaluated deterministic curriculum learning strategies on small TSP instances. For space reasons, our discussion below emphasizes the approaches using deep RL as a constructive heuristic (which builds a solution iteratively).

Invariance and equivariance have been important properties to exploit for designing powerful deep learning architectures (Bengio & Lecun, 1997; Gens & Domingos, 2014; Cohen & Welling, 2016). They have also inspired some of the previous work on solving TSP problems, although they were often not explicitly discussed like in our work. For instance, Deudon et al. (2018) use principal component analysis to exploit rotation invariance as a single preprocessing step. In contrast to their work, we compose various preprocessing transformations to be applied at every solving step. To the best of our knowledge, we are the first to propose such technique. In addition, permutation invariance is the motivation for using attention models (Kool et al., 2019; Deudon et al., 2018) or GNNs (Ma et al., 2019) in RL solvers. Contrary to Deudon et al. (2018), Kool et al. (2019) select the next city to visit based on the first and last visited cities while Ma et al. (2019) propose to use relative positions for translation invariance. Our proposition combines those two ideas, but compared to Kool et al. (2019), our model is based on GNN, while compared to Ma et al. (2019), it has a simpler architecture that does not require an LSTM. For vehicle routing problems, Peng et al. (2020) propose to remove visited cities from the RL agent's input, a technique we also apply to TSP. In contrast to their work, our GNN model is simpler and we investigate generalization from small instances to very large instances. To summarize, compared to all previous work, our paper investigates in a more systematic fashion the exploitation of equivariance to help training and improve generalization.

Several previous studies combine deep learning or RL with various local search techniques to find better solutions: 2-opt (Deudon et al., 2018; Ma et al., 2019; Da Costa et al., 2020; Wu et al., 2021b), $k$-opt (Zheng et al., 2021), or Monte-Carlo tree search (Fu et al., 2021). In contrast to these methods, we use a combined local search technique, which applies several heuristics in an efficient way. Moreover, to the best of our knowledge, no previous work considers interleaving local search with policy gradient updates to obtain a more synergetic final method for solving TSP. For the bin packing problem, Cai et al. (2019) use RL to learn a perturbative heuristics to provide a good initial

solution to a heuristic optimizer. Our interleaved training process is based on a similar idea, but our RL agent learns a constructive heuristics and we provide an intuitive motivation why this approach may work via our smoothed policy gradient.

All these machine-learning-based methods can be categorized as learning from a batch of instances or as directly learning on the instance to be solved. Like our work, most methods are part of the first category. However, some recent work belongs to the second category (Zheng et al., 2021) or are hybrid (Fu et al., 2021). For instance, Zheng et al. (2021) apply RL to make the LKH algorithm (Helsgaun, 2017), a classic efficient heuristic for TSP, adaptive to the instance to be solved. In Fu et al. (2021)'s method, evaluations learned in a supervised way from a batch of instances are used in Monte-Carlo tree search so that solutions can adaptively be found for the current instance. Being adaptive to the current instance can undoubtedly boost the performance of the solver. We leave for future work the improvement of our solver to become more adaptive to the instance to be solved.

## 3  PRELIMINARIES

Following previous work, we focus on the symmetric 2D Euclidean TSP, which we recall below. We then explain how a TSP instance can be tackled with RL. We first provide some notations:

For any positive integer $N \in \mathbb{N}$, $[N]$ denotes the set $\{1, 2, \ldots, N\}$. Vectors and matrices are denoted in bold (e.g., $\boldsymbol{x}$ or $\boldsymbol{X}$). For a set of subscripts $I$, $\boldsymbol{X}_I$ denotes the matrix formed by the rows of $\boldsymbol{X}$ whose indices are in $I$.

**Traveling Salesman Problem**   A symmetric 2D Euclidean TSP instance is described by a set of $N$ cities (identified to the set $[N]$) and their coordinates in $\mathbb{R}^2$. The goal in this problem is to find the shortest tour that visits each city exactly once. The coordinates of city $i$ are denoted $\boldsymbol{x}_i \in \mathbb{R}^2$. The matrix whose rows correspond to city coordinates is denoted $\boldsymbol{X} \in \mathbb{R}^{N \times 2}$. A feasible solution (i.e., tour) of a TSP instance is a permutation $\sigma$ over $[N]$ with length equal to:

$$L_\sigma(\boldsymbol{X}) = \sum_{t=1}^{N} \|\boldsymbol{x}_{\sigma(t)} - \boldsymbol{x}_{\sigma(t+1)}\|_2 \tag{1}$$

where $\|\cdot\|_2$ denotes the $\ell_2$-norm, $\sigma(t) \in [N]$ is the $t$-th visited city in tour $\sigma$, and by abuse of notation, $\sigma(N+1)$ denotes $\sigma(1)$. Therefore, solving a TSP instance consists in finding the permutation $\sigma$ that minimizes the tour length $L_\sigma(\boldsymbol{X})$ defined in Equation (1). Since TSP solutions are invariant to scaling, we assume that the city coordinates are in the square $[0,1]^2$. In Appendix A, we recall some heuristics (e.g., insertion, $k$-opt) that have been proposed for solving TSP.

**RL as a Constructive Heuristic**   RL can be used to construct a TSP tour $\sigma$ sequentially. Intuitively, at iteration $t \in [N]$, an RL solver (i.e., policy) selects the next unvisited city $\sigma(t)$ to visit based on the current partial tour and the description of the TSP instance (i.e., coordinates of cities). Therefore, this RL problem corresponds to a repeated $N$-horizon sequential decision-making problem. As noticed by Kool et al. (2019), the decision for the next city to visit only depends on the description of the TSP instance and the first and last visited cities. In addition, we update the description of the TSP instance by removing the coordinates of visited cities. Surprisingly, to the best of our knowledge, no previous work exploits this simplified RL model.

Formally, in the RL language, at time step $t \in [N]$, an action $a_t$ represents the next city to visit, i.e., $a_t = \sigma(t)$. Let $I_1 = [N]$ and $I_{t+1} = [N] \backslash \{\sigma(1), \ldots, \sigma(t)\}$ be the set of remaining cities after $t$ cities have been already visited. Moreover, let $J_1 = [N]$ and $J_t = I_t \cup \{\sigma(1), \sigma(t)\}$ be the set of unvisited cities in addition of the first and last visited cities ($\sigma(1)$ and $\sigma(t)$). Therefore, $a_t \in I_t$ for all $t \in [N]$. A state $\boldsymbol{s}_t$ can be represented by a matrix $\boldsymbol{X}_{J_t}$ with flags indicating the first and last visited cities. This matrix includes the coordinates of unvisited cities ($\boldsymbol{X}_{I_t}$) in addition to the coordinates of the first and last visited cities ($\boldsymbol{x}_{\sigma(1)}$ and $\boldsymbol{x}_{\sigma(t)}$). Note that at $t = 1$, no city has been chosen yet, so the initial state $\boldsymbol{s}_1$ only contains the list of city coordinates. A state $\boldsymbol{s}_N$ contains the coordinates of the unvisited cities and those of the first and last visited cities. After choosing the last city to visit in state $\boldsymbol{s}_N$, the tour $\sigma$ is completely generated. The immediate reward $\boldsymbol{r}(\boldsymbol{s}_t, a_t)$ for an action $a_t$ in a state $\boldsymbol{s}_t$ can be defined as the negative length between the last visited city and the next

chosen city, since we want the tour length to be small:

$$
r\left(\boldsymbol{s}_t, a_t\right) = \begin{cases} 0 & \text{for } t = 1 \\ -||\boldsymbol{x}_{\sigma(t)} - \boldsymbol{x}_{\sigma(t-1)}||_2 & \text{for } t = 2, \dots, N-1 \\ -||\boldsymbol{x}_{\sigma(N)} - \boldsymbol{x}_{\sigma(N-1)}||_2 - ||\boldsymbol{x}_{\sigma(1)} - \boldsymbol{x}_{\sigma(N)}||_2 & \text{for } t = N \end{cases} \quad (2)
$$

After choosing the first city $a_1 = \sigma(1)$, the reward is zero since no length can be computed. After the final action $a_N$, an additional reward $-||\boldsymbol{x}_{\sigma(1)} - \boldsymbol{x}_{\sigma(N)}||_2$ is added to complete the tour length.

In deep RL, a policy $\pi_{\boldsymbol{\theta}}$ is represented as a neural network parametrized by $\boldsymbol{\theta}$. The goal is then to find $\boldsymbol{\theta}^*$ that maximizes the objective function $J(\boldsymbol{\theta})$:

$$
\boldsymbol{\theta}^\star = \arg\max_{\boldsymbol{\theta}} J(\boldsymbol{\theta}) = \arg\max_{\boldsymbol{\theta}} \mathbb{E}_{\tau \sim p_{\boldsymbol{\theta}}(\tau)} \left[ \sum_{t=1}^{N} r_t \right], \quad (3)
$$

where for all $t \in [N]$, $r_t = r(\boldsymbol{s}_t, a_t)$, $\tau = (\boldsymbol{s}_1, a_1, \boldsymbol{s}_2, a_2, \dots, \boldsymbol{s}_N, a_N)$ is a complete trajectory, and $p_{\boldsymbol{\theta}}$ is the probability over trajectories induced by policy $\pi_{\boldsymbol{\theta}}$. This objective function $J(\boldsymbol{\theta})$ can be optimized by a policy gradient (Williams, 1992) or actor-critic method (Sutton & Barto, 1998).

## 4 EQUIVARIANT MODEL

In this section, we present several techniques to exploit invariances and equivariances in the design of an RL solver. Formally, a mapping $f : A \to B$ from a set $A$ to a set $B$ is invariant with respect to a symmetry $\rho^A : A \to A$ iff $f(x) = f(\rho^A(x))$ for any $x \in A$. More generally, given a symmetry that acts on $A$ with $\rho^A : A \to A$ and on $B$ with $\rho^B : B \to B$, a mapping $f : A \to B$ is equivariant with respect to this symmetry iff $\rho^B(f(x)) = f(\rho^A(x))$ for any $x \in A$. This definition shows that invariance is a special case of equivariance when $\rho^B$ is the identity function. Intuitively, equivariance for an RL solver (resp. its value function) means that if a transformation is applied to its input, its output (resp. its value) can be recovered by a corresponding transformation.

In the remaining of the paper, for simplicity, we often use *equivariance* to refer to both equivariance and invariance, since the latter is a special case of the former. Equivariance can be exploited in RL in various ways. We consider some equivariant preprocessing methods on the description of TSP instances to standardize the kinds of instances the solver is trained on and evaluated on. In addition, we propose a simple deep learning model for which we apply some other equivariant preprocessing methods on its inputs to further reduce the input space.

### 4.1 EQUIVARIANT PREPROCESSING OF TSP INSTANCE

An RL solver should be invariant with respect to any Euclidean symmetry (rotation, reflection, translations and their compositions) and to any positive scaling transformation applied on city positions. To enforce these invariances, we can apply these transformations to preprocess the inputs of the solver such that the transformed inputs are always in a standard form. Doing so allows the solver to be trained on more similar inputs.

Concretely, for rotation invariance, we rotate and scale the city positions such that they are mostly distributed along the first diagonal of the square $[0, 1]^2$. This can be achieved by performing a principal component analysis, rotating the first found axis by $45°$ anti-clockwise and scaling to fit the cities in the $[0, 1]^2$ square. This transformation, which is slightly different from (Deudon et al., 2018), allows the cities to be as spread as possible in $[0, 1]^2$. For scaling and translation invariance, we apply a scaling and translation transformation to the city positions such that there are a maximum number of cities (i.e., 2 or 3 depending on configuration) on the border of the square $[0, 1]^2$. For reflections, we only consider horizontal, vertical, and diagonal flips (with respect to the coordinate axes) for simplicity. Cities are reflected such that a majority of them are in a fixed chosen region.

Since the symmetries can be composed, these preprocessing methods can be applied sequentially. Theoretically, it would be beneficial to apply all of them in combination, however this has a computational cost. It is therefore more effective to only use a selection of the most effective ones. Moreover, these preprocessing methods can be applied either once on the initial TSP instance ($\boldsymbol{X}$), or at each solving step $t$ on the remaining cities ($\boldsymbol{X}_{I_t}$). Theoretically, performing these preprocessings iteratively should help most, since the RL solver is only trained on standardized inputs.

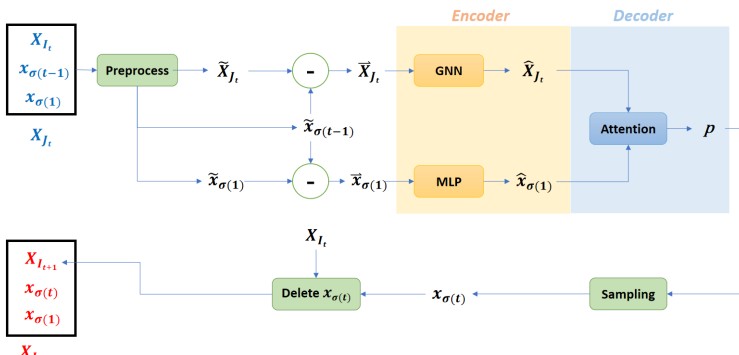

Figure 1: Model Architecture of eMAGIC

This is confirmed by our experimental analysis. We find out that the combination that provides the best results are rotation followed by scaling and translation, which will be used in the experimental evaluation of our method. However, our overall approach is generic and could include any other symmetries for which equivariance would hold for the RL solver. In Appendix F, we present the details of the evaluation of the different preprocessing methods.

In the remaining, we assume that the set of equivariant transformations is fixed. For any set of indices $I \subseteq [N]$, we denote the matrix of positions $X_I$ after preprocessing by $\widetilde{X}_I$ and any coordinates $x$ after preprocessing by $\tilde{x}$. Note that the preprocessing step (e.g., if it includes the rotation transformation described above) may depend on the initial matrix $X_I$, but we prefer not to reflect it in the notations to keep them simple.

## 4.2 EQUIVARIANT MODEL

Our proposed model, which represents the RL solver, has an encoder-decoder architecture (see Figure 1). In our model, the first city to visit is fixed arbitrarily, since the construction of the tour should be invariant to the starting city. Therefore, the decisions of the solver starts at time step $t \geq 2$. Our model is designed to take into account other equivariances that are known to hold in the problem. Invariance with respect to translation can be further exploited by considering relative positions with respect to the last visited city instead of the original absolute positions, as suggested by Ma et al. (2019). Therefore, we define the input of our RL solver to only include two pieces of information. First, the information about the current partial tour now only needs the relative preprocessed position of the first visited city ($\vec{x}_{\sigma(1)} = \widetilde{x}_{\sigma(1)} - \widetilde{x}_{\sigma(t)}$). Since the last visited city is always represented by the origin, it can be dropped. Second, the information about the remaining TSP instance corresponds to the relative preprocessed positions of the unvisited cities ($\vec{x}_{\sigma(n)} = \widetilde{x}_{\sigma(n)} - \widetilde{x}_{\sigma(t-1)}$ for $n \in I_t$) and of the first and last visited cities ($\vec{x}_{\sigma(1)}$ and the origin $\vec{x}_{\sigma(t)} = \mathbf{0}$). We denote the matrix of those relative preprocessed positions by $\overrightarrow{X}_{J_t}$, i.e., the matrix whose rows are the rows of $\widetilde{X}_{J_t}$ (obtained from $X_{J_t}$) minus $\widetilde{x}_{\sigma(t)}$. The last visited city needs to be kept in this matrix, since it represents the labels of the nodes of the remaining graph on which the tour should be completed.

**Encoder** With $H$ denoting the embedding dimension, the encoder of our model is composed of a graph neural network (GNN) (Battaglia et al., 2018), which computes an embedding $\widehat{X}_{J_t} \in \mathbb{R}^{|J_t| \times H}$ of the relative city positions $\overrightarrow{X}_{J_t} \in \mathbb{R}^{|J_t| \times 2}$, and a multilayer perceptron (MLP), which computes an embedding $\hat{x}_{\sigma(1)} \in \mathbb{R}^H$ of the relative position of the first visited city $\overrightarrow{x}_{\sigma(1)} \in \mathbb{R}^2$. The GNN encodes the information about the graph describing the remaining TSP problem. As a GNN, it is equivariant with respect to the order of its inputs and its outputs depend on the graph structure and information (i.e., city positions). The MLP encodes the information about the first city. Since we exploit the independence with respect to the visited cities between the first and last cities (not included), our model does not require any complex architecture, like LSTM (Hochreiter & Schmidhuber, 1997) or GNN, for encoding the positions of the visited cities, in contrast to most previous work. A simple MLP suffices since only the relative position of the first city is required.

Formally, the GNN computes its output $\widehat{\boldsymbol{X}}_{J_t} \in \mathbb{R}^{|J_t| \times H}$ from inputs $\overrightarrow{\boldsymbol{X}}_{J_t} \in \mathbb{R}^{|J_t| \times 2}$ through $n_{\text{GNN}}$ layers, which allows each city to get information from its neighbors up to $n_{\text{GNN}}$ edges away:

$$\boldsymbol{X}^{(0)} = \overrightarrow{\boldsymbol{X}} \boldsymbol{\Theta}^{(0)} \tag{4}$$

$$\boldsymbol{X}^{(\ell)} = \lambda \cdot \boldsymbol{X}^{(\ell-1)} \boldsymbol{\Theta}^{(\ell)} + (1 - \lambda) \cdot \mathrm{F}^{(\ell)} \left( \frac{\boldsymbol{X}^{(\ell-1)}}{|J_t| - 1} \right) \tag{5}$$

where $\boldsymbol{\Theta}^{(0)} \in \mathbb{R}^{2 \times H}$ and $\boldsymbol{\Theta}^{(\ell)} \in \mathbb{R}^{H \times H}$ are learnable weights, $\boldsymbol{X}^{(\ell-1)} \in \mathbb{R}^{|J_t| \times H}$ is the input of the $\ell^{th}$ layer of the GNN for $\ell \in [n_{\text{GNN}}]$, $\boldsymbol{X}^{(n_{\text{GNN}})} = \widehat{\boldsymbol{X}}_{J_t}$, $\mathrm{F}^{(\ell)} : \mathbb{R}^{|J_t| \times H} \to \mathbb{R}^{|J_t| \times H}$ is the aggregation function, which is implemented as a neural network, and $\lambda \in [0, 1]$ is another trainable parameter. Visual illustration of the GNN is deferred to Appendix H for space reasons.

**Decoder**   Once the embeddings $\widehat{\boldsymbol{X}}_{J_t}$ and $\widehat{\boldsymbol{x}}_{\sigma(0)}$ are computed, the probability of selecting a next city to visit is obtained via a standard attention mechanism (Bello et al., 2016) using $\widehat{\boldsymbol{X}}_{J_t}$ and $\widehat{\boldsymbol{x}}_{\sigma(0)}$ as the keys and query respectively. Formally, the decoder outputs a vector $\boldsymbol{u} \in \mathbb{R}^N$ expressed as:

$$\boldsymbol{u}_j = \begin{cases} -\infty & \forall j \in I_t \\ \boldsymbol{w} \cdot \tanh \left( \widehat{\boldsymbol{X}}_{J_t, j} \boldsymbol{\Theta}_g + \boldsymbol{\Theta}_m \widehat{\boldsymbol{x}} \right) & \text{otherwise} \end{cases} \tag{6}$$

where $\boldsymbol{u}_j$ is the $j^{th}$ entry of the vector $\boldsymbol{u}$, $\widehat{\boldsymbol{X}}_{J_t, j}$ is the $j^{th}$ row of the matrix $\widehat{\boldsymbol{X}}_{J_t}$, $\boldsymbol{\Theta}_g$ and $\boldsymbol{\Theta}_m$ are trainable matrices with shape $H \times H$, $\boldsymbol{w} \in \mathbb{R}^N$ is another trainable weight vector. Then, a softmax transformation turns $\boldsymbol{u}$ into a probability distribution $\boldsymbol{p} = (\boldsymbol{p}_j)_{j \in [N]}$ over unvisited cities:

$$\boldsymbol{p} = \text{softmax}(\boldsymbol{u}) = \left( \frac{e^{\boldsymbol{u}_j}}{\sum_{j=1}^N e^{\boldsymbol{u}_j}} \right)_{j \in [N]} \tag{7}$$

where $\boldsymbol{p}_j$ is the $j^{th}$ entry of the probability distribution $\boldsymbol{p}$ and $\boldsymbol{u}_j$ is the $j^{th}$ entry of the vector $\boldsymbol{u}$. Note that the probability of any visited city $j \in I_t$ is zero since $\boldsymbol{u}_j = -\infty$ in Equation (6).

## 5   ALGORITHM & TRAINING

We introduce three innovative training techniques that we apply during our training process: combined local search, smoothed policy gradient, and stochastic curriculum learning. Overall, these techniques can help train a fast and generalizable policy, which is able to generate tours that can easily be improved by local search. The overall algorithm[1] can be found in appendix C.

**Combined Local Search**   In contrast to previous work, which generally only considers 2-opt, we propose to use a combination of several (possibly random) local search methods (i.e., local insertion heuristic, random 2-opt, search 2-opt and search random 3-opt) to improve a tour generated by the RL solver. For space reasons, we defer their presentations in Appendix A. A local search method can heuristically improve a tour, but may get stuck in a local optimum. Using a combination of them alleviates this issue, since different heuristics usually have different local optima. Another important distinction with previous work is that we also use local search during training, not only testing.

**Smoothed Policy Gradient**   For simplicity, we train our model with the REINFORCE algorithm (Williams, 1992), which leverages the policy gradient for optimization. However, instead of using the standard policy gradient, which is based on the value of the tour generated by the policy, we compute the policy gradient with the value of the tour improved by our combined local search. While standard RL training may yield policies whose outputs may not easily be improved by local search, our new definition directly trains the RL solver to find solutions that can be improved by local search, which allows RL and local search to have synergetic effects.

Intuitively, this new policy gradient amounts to smoothing the objective function $J(\boldsymbol{\theta})$ that is optimized. Recall $J(\boldsymbol{\theta}) = -\mathbb{E}[L_\sigma(\boldsymbol{X})]$, where the expectation is taken with respect to $\sigma$, which is a

---

[1]Our implementation takes advantage of GPU acceleration when possible. The source code will be shared after publication.

random variable corresponding to the tour generated by policy $\pi_{\boldsymbol{\theta}}$. In our approach, this usual objective function is replaced by $J^+(\boldsymbol{\theta}) = -\mathbb{E}[L_{\sigma_+}(\boldsymbol{X})]$, where $\sigma_+$ is a random variable corresponding to the improved tour obtained by our combined local search from $\sigma$. Therefore, this last expectation is taken with respect to the probability distributions generated by policy $\pi_{\boldsymbol{\theta}}$ and our combined local search. This objective function can be understood as $J^+(\boldsymbol{\theta}) = -\mathbb{E}[\min_{\sigma' \in \mathcal{N}(\sigma)} L_{\sigma'}(\boldsymbol{X})]$ where $\mathcal{N}(\sigma)$ is a neighborhood of $\sigma$ defined by local search. This stochastic $\min$ operation has a smoothing effect on $J(\boldsymbol{\theta})$. That is why, we call the gradient of $J^+(\boldsymbol{\theta})$ a *smoothed policy gradient*.

Formally, this novel policy gradient can be estimated on a batch of TSP instances $\boldsymbol{X}^{(1)}, \ldots, \boldsymbol{X}^{(B)}$:

$$\nabla_{\boldsymbol{\theta}} J^+(\boldsymbol{\theta}) \approx -\frac{1}{|B|} \sum_{b=1}^{|B|} \left( \sum_{t=2}^{N} \nabla_{\boldsymbol{\theta}} \log \pi_{\boldsymbol{\theta}}(a_t^{(b)} | \boldsymbol{s}_t^{(b)}) \right) \left( L_{\sigma_+^{(b)}}(\boldsymbol{X}^{(b)}) - l^{(b)} \right), \qquad (8)$$

where $\sigma^{(b)}$ is the tour generated by the current policy $\pi_{\boldsymbol{\theta}}$ on instance $\boldsymbol{X}^{(b)}$, $\sigma_+^{(b)}$ is the improved tour obtained by our combined local search starting from $\sigma^{(b)}$, and $l^{(b)} = -L_{\sigma^{(b)}}(\boldsymbol{X}^{(b)})$ is a novel baseline, which we call *policy rollout* baseline, used to reduce the variance of the policy gradient estimation. It enjoys the nice property that it does not require additional calculations, since $L_{\sigma^{(b)}}(\boldsymbol{X}^{(b)})$ is computed when $\sigma^{(b)}$ is generated. In our experiments, the policy rollout baseline easily outperforms the previous greedy baselines (Ma et al., 2019; Kool et al., 2019).

Note that by construction, $L_{\sigma_+^{(b)}}(\boldsymbol{X}^{(b)}) \leq L_{\sigma^{(b)}}(\boldsymbol{X}^{(b)}) = -\sum_{t=1}^{N} r_t^{(b)}$. Therefore, the smoothed policy gradient updates more if our combined local search can make more improvements upon this certain policy. For completeness, we provide more details in Appendix B.

**Stochastic Curriculum Learning**   Curriculum Learning (CL) is widely used in machine learning (Soviany et al., 2021). Its basic principle is to control the increase of the difficulty of training instances. CL can speed up learning and improve generalization (Weinshall et al., 2018). We apply *stochastic* CL to train our model. Instead of a deterministic process, stochastic CL increases the difficulty according to a probability distribution. We choose the TSP size (i.e., number of cities) as a measure of difficulty for a TSP instance. We assume it to be in the range of $\mathcal{R} = \{10, 11, \ldots, 50\}$. For each epoch $e$, we define the vector $\boldsymbol{g}^{(e)} \in \mathbb{R}^{41}$ (since $|\mathcal{R}| = 41$) to be:

$$\boldsymbol{g}_k^{(e)} = \frac{1}{\sqrt{2\pi}\sigma_N} \exp^{-\frac{1}{2}\left(\frac{(k+10)-e}{\sigma_N}\right)^2}, \qquad (9)$$

where $\boldsymbol{g}_k^{(e)}$ is the $k$-th entry of $\boldsymbol{g}^{(e)}$ and hyperparameter $\sigma_N$ is the standard deviation of this Gaussian density function. Then, $\boldsymbol{g}^{(e)}$ is turned into a categorical distribution $\boldsymbol{p}^{(e)} \in [0, 1]^{41}$ via a softmax:

$$\boldsymbol{p}^{(e)} = \text{softmax}(\boldsymbol{g}^{(e)}) \qquad (10)$$

The $k$-th entry of $\boldsymbol{p}^{(e)}$ gives the probability of choosing a TSP instance of size $(k + 10)$ at epoch $e$.

## 6   EXPERIMENTAL RESULTS

We present three sets of experiments. First, to validate the effectiveness of eMAGIC, we train our model on randomly-generated TSP instances (using stochastic CL with sizes up to 50), and test the model on other randomly-generated TSP instances (TSP$n$ where size $n = 20$ up to 1000). Second, to further prove of its generalization capability, we directly evaluate models trained on random instances on realistic symmetric 2D Euclidean TSP instances with sizes range from 51 to 1002 in TSPLIB (Reinelt, 1991). Third, we conduct an ablation study to show the significance of every component of eMAGIC (i.e., equivariance, stochastic CL, policy rollout baseline, combined local search, and RL). We evaluate four versions of our model: eMAGIC(G), eMAGIC(S), eMAGIC(s1) and eMAGIC(s10) where G means the tour is generated greedily from the RL policy, while the other ones are based on random sampling and differ with respect to the number of times eMAGIC is applied (once for s1, 10 times for s10 and 100 for S). The details about the experimental settings and the used hyperparameters, which are the same for all experiments, are provided in Appendix D.

**Performance on Randomly Generated TSP**   We compare the performance of our model with 12 other methods in Tables 1 and 2, which covers various types of TSP solvers including exact solvers,

Table 1: Results of eMAGIC vs baselines, tested on 10,000 instances for TSP 20, 50 and 100.

| Method | Type[†] | TSP20 | | | TSP50 | | | TSP100 | | |
|---|---|---|---|---|---|---|---|---|---|---|
| | | Len. | Gap | Time | Len. | Gap | Time | Len. | Gap | Time |
| Concorde[*] | ES | 3.830 | 0.00% | 2.3m | 5.691 | 0.00% | 13m | 7.761 | 0.00% | 1.0h |
| Gurobi[*] | ES | 3.830 | 0.00% | 2.3m | 5.691 | 0.00% | 26m | 7.761 | 0.00% | 3.6h |
| LKH3[*] | H | 3.830 | 0.00% | 21m | 5.691 | 0.00% | 27m | 7.761 | 0.00% | 50m |
| 2-opt | H | 4.082 | 6.56% | 0.3s | 6.444 | 13.24% | 2.3s | 9.100 | 17.26% | 9.3s |
| Random[♯] | H | 4.005 | 4.57% | 3.3m | 6.128 | 7.69% | 12m | 8.511 | 9.66% | 17m |
| Farthest[♯] | H | 3.932 | 2.64% | 4.0m | 6.010 | 5.62% | 10m | 8.360 | 7.71% | 21m |
| GCN[*1] | SL(G) | 3.855 | 0.65% | 19s | 5.893 | 3.56% | 2.0m | 8.413 | 8.40% | 11m |
| GCN[*1] | SL(BS) | 3.835 | 0.12% | 21m | 5.707 | 0.29% | 35m | 7.876 | 1.48% | 32m |
| GCN[*1] | SL(BST) | 3.831 | 0.01% | 22m | 5.692 | 0.03% | 38m | 7.872 | 1.43% | 1.2h |
| AGCRN+M[*2] | SL+M | **3.830** | **0.00%** | 1.6m | **5.691** | **0.01%** | 7.9m | 7.764 | 0.04% | 15m |
| GAT[*3] | RL(S) | 3.874 | 1.14% | 10m | 6.109 | 7.34% | 20m | 8.837 | 13.87% | 48m |
| AM[*4] | RL(G) | 3.841 | 0.29% | 6.0s | 5.785 | 1.66% | 34s | 8.101 | 4.38% | 1.8m |
| AM[*4] | RL(S) | 3.832 | 0.05% | 17m | 5.719 | 0.49% | 23m | 7.974 | 2.74% | 1.2h |
| GPN[5] | RL | 4.074 | 6.35% | 0.8s | 6.059 | 6.47% | 2.5s | 8.885 | 14.49% | 6.2s |
| **eMAGIC(G)** | RL(LS) | 3.841 | 0.29% | 2.8s | 5.732 | 0.74% | 16s | 7.923 | 2.09% | 1.4m |
| **eMAGIC(S)** | RL(LS) | **3.830** | **0.00%** | 38s | **5.691** | **0.01%** | 3.5m | **7.762** | **0.02%** | 14.6m |

[*] as reported in previous work. [♯] Random - Random Insertion; Farthest - Farthest Insertion.
[1] (Joshi et al., 2019a), [2] (Fu et al., 2021), [3] (Deudon et al., 2018), [4] (Kool et al., 2019), [5] (Ma et al., 2019)
[†] ES - Exact Solver; H - Heuristic; SL - Supervised Learning; RL - Reinforcement Learning; G - Greedy; S - Sampling; M - Monte-Carlo Tree Search; LS - Combined Local Search; BS - Beam Search; BST - Beam Search and Shortest Tour Heuristic.

Table 2: Results of eMAGIC vs baselines, tested on 10,000 instances for TSP 200, 500, and 1000.

| Method | Type[†] | TSP200 | | | TSP500 | | | TSP1000 | | |
|---|---|---|---|---|---|---|---|---|---|---|
| | | Len. | Gap | Time | Len. | Gap | Time | Len. | Gap | Time |
| Concorde[*] | ES | 10.72 | 0.00% | 3.4m | 16.55 | 0.00% | 38m | 23.12 | 0.00% | 7h |
| Gurobi[*] | ES | - | - | - | - | - | - | - | - | - |
| LKH3[*] | H | 10.72 | 0.00% | 2.0m | 16.55 | 0.00% | 11m | 23.12 | 0.00% | 38m |
| 2-opt | H | 12.84 | 19.80% | 34s | 20.44 | 23.51% | 3.3m | 28.95 | 25.23% | 14m |
| Random[♯] | H | 11.84 | 10.47% | 27s | 18.59 | 12.34% | 1.1m | 26.12 | 12.98% | 2.3m |
| Farthest[♯] | H | 11.64 | 8.63% | 33s | 18.31 | 10.64% | 1.4m | 25.74 | 11.35% | 2.9m |
| GCN[*1] | SL(G) | 17.01 | 58.73% | 59s | 29.72 | 79.61% | 7m | 48.62 | 110.3% | 29m |
| GCN[*1] | SL(BS) | 16.19 | 51.02% | 4.6m | 30.37 | 83.55% | 38m | 51.26 | 122% | 52m |
| GCN[*1] | SL(BST) | 16.21 | 51.21% | 4.0m | 30.43 | 83.89% | 31m | 51.10 | 121% | 3.2h |
| AGCRN+M[*2] | SL+M | 10.81 | 0.88% | 2.5m | **16.97** | **2.54%** | 5.9m | **23.86** | **3.22%** | 12m |
| GAT[*3] | RL(S) | 13.18 | 22.91% | 4.8m | 28.63 | 73.03% | 20m | 50.30 | 117.6% | 38m |
| AM[*4] | RL(G) | 11.61 | 8.31% | 5.0s | 20.02 | 20.99% | 1.5m | 31.15 | 34.75% | 3.2m |
| AM[*4] | RL(S) | 11.45 | 6.82% | 4.5m | 22.64 | 36.84% | 16m | 42.80 | 85.15% | 1.1h |
| GPN[*5] | RL | - | - | - | 19.61 | 18.49% | - | 28.47 | 23.15% | - |
| GPN+2opt[*5] | RL+2opt | - | - | - | 18.36 | 10.95% | - | 26.13 | 13.02% | - |
| GPN[5] | RL | 13.28 | 23.87% | 2.5s | 23.64 | 42.87% | 7.1s | 37.85 | 63.72% | 18s |
| **eMAGIC(G)** | RL(LS) | 11.14 | 3.89% | 36s | 17.52 | 5.89% | 2.0m | 24.70 | 6.85% | 4.9m |
| **eMAGIC(S)** | RL(LS) | **10.77** | **0.50%** | 2.4m | 17.03 | 2.92% | 9.7m | 24.13 | 4.36% | 27m |

See footnotes of Table 1.

heuristics, and learning-based approaches. Column 1 and 2 of both Tables 1 and 2 correspond to the method name and the method type, respectively. Columns 3, 4, and 5 provide the average tour length, gap to the optimal (provided by Concorde (Applegate et al., 2004)), and computational time, respectively. For space reasons, we only include results of eMAGIC(G) and eMAGIC(S) in Tables 1

Table 3: Gap to optimal for different ranges of instances in TSPLIB.

| Size Range | eMAGIC(S) | Wu et al.[*1] | S2VDQN[*2] | OR[3] | AM[*4] | L2OPT[*5] | Furthest[6] |
|---|---|---|---|---|---|---|---|
| $50-199$ | **0.46%** | 15.00% | 3.77% | 3.49% | 78.73% | 6.54% | 7.60% |
| $200-399$ | **1.37%** | 23.49% | 6.87% | 3.61% | 293.76% | 12.17% | 9.54% |
| $400-1002$ | **3.40%** | - | - | 3.57% | - | - | 10.11% |

[*] as reported in previous work.
[1] (Wu et al., 2021a), [2] (Khalil et al., 2017), [3] (Perron & Furnon), [4] (Kool et al., 2019), [5] (de O. da Costa et al., 2020), [6] (Furthest Insertion).

and 2, and leave results of other versions in Table 4 in Appendix E.1. Moreover, we provide the result variances for our methods and evaluate them on TSP10,000 in resp. Appendices E.2 and E.3.

Tables 1 and 2 show that the computational times of exact solvers become prohibitive as the TSP size increases. Note that Gurobi is not able to solve TSP instances larger than 200 under a reasonable time budget. The classic heuristic methods are relatively fast, but their performances are not satisfactory. Among all learning methods, Fu et al. (2021) provide excellent results, but it is based on Monte-Carlo tree search, which adapts to the instance to be solved. Without this adaptivity, our model with or without sampling provides competitive results. It can be better than Fu et al. (2021) up to TSP200. This somewhat suggests the limit of our approach, which trains on small instances and directly generalizes to large ones, without learning on the instance to be solved. We expect that our approach could be improved by training on slightly bigger instances (e.g., up to 100) or adding an adaptive component like in Fu et al. (2021).

**Performance on Realistic TSP**   Table 3 compares the performances of a variety of learning-based TSP solvers on instances from TSPLIB. Each column of Table 3 represents the average gap to the optimal solution over the instances indicated by the corresponding rows. We treat the tested instances as small instances if their sizes are under 200 and large instances otherwise. And we extend the testing to larger instances with size up to 1002 and leave the performances of other models as empty for size ranging from 400 to 1002 since in their papers, the testings stop at instances with sizes around 400. We can observe from Table 3 that our model can perform much better than the other learning-based solvers not only for small problems but also large ones, which demonstrates the strong ability and the practical significance of eMAGIC to tackle realistic TSP problems. When the testing size increases, most models suffer from a relatively big increasing of the average gap while ours only increases less than $1\%$ and remains in a good absolute value ($3.40\%$) for even larger instances, which indicates a strong generalization ability of our model. Further details and experimental results are provided in Appendix I.

**Ablation Study**   We demonstrate the strength of all the techniques we applied (including equivariance, stochastic CL, policy rollout baseline, combined local search, and RL) using an ablation study. We turn off each feature one at a time to see if the performance drops compared to the full version of eMAGIC(s1). For space reasons, details of the ablation study are deferred to Appendix G. Table 10 and 11 demonstrate that each technique plays a role in our model.

## 7   CONCLUSION

We presented a combination of novel techniques (notably, equivariance, combined local search, smoothed policy gradient) for designing an RL solver for TSP, which shows a good generalization capability. We demonstrated its effectiveness both on random and realistic instances, which shows that our model can reach state-of-the-art performance.

For future work, the approach can be further improved in various ways, e.g., extending it to the actor-critic scheme and exploiting invariances with the critic; or making it adaptive and learn on the instance to be solved. The approach could also be applied on other combinatorial optimization problems and other RL problems.

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

# A    COMBINED LOCAL SEARCH

We first provide a brief introduction to TSP heuristics and then give the detailed presentation of each local search method we used, together with our whole combined local search algorithm.

Since TSP is an NP-hard problem (Papadimitriou, 1977), solving exactly large TSP instances is generally impractical. Hence, many heuristics have been proposed to solve TSP. Two important categories of heuristics are constructive heuristics and local search heuristics. On the one hand, *constructive* heuristics iteratively build a tour from scratch and therefore, do not rely on existing tours. As an example, *insertion heuristics* are constructive: they first choose a starting city randomly, then repeatedly insert an unvisited city into the partial tour that minimizes the increase of tour length until all the cities are included in the tour. The unvisited city can be selected in different ways leading to various versions of insertion heuristics: random insertion, nearest insertion, or farthest insertion (see Kool et al. (2019) for implementation details). Among them, farthest insertion usually yields the best results. On the other hand, *local search* heuristics try to improve a given complete tour by perturbing it. They are widely used as post optimization for TSP solvers, such as Deudon et al. (2018); Ma et al. (2019). One important class of local search heuristic is *k-opt*, which improves an existing complete tour $\sigma$ by repeatedly performing the following operation: remove $k$ edges of the current tour and reconnect the obtained subtours in order to decrease the tour length. For instance, one 2-opt operation would replace:

$$\sigma = \big(\sigma(1), \sigma(2)..., \sigma(i), ..., \sigma(j), ..., \sigma(N)\big)$$

by

$$\sigma' = \big(\sigma(1), ..., \sigma(i), \sigma(j), \sigma(j-1), ..., \sigma(i+1), \sigma(j+1), ..., \sigma(N)\big)$$

where $i < j < N$ if $L_{\sigma'}(X) < L_{\sigma}(X)$. Based on $k$-opt, the LKH algorithm (Helsgaun, 2017) can often achieve nearly optimal solutions, but requires a long runtime.

Based on how many edges are removed and how the subtours are reconnected during one step, every local search heuristic can be seen as a special case of $k$-opt. Since the number of possible $k$-opt operations is $\mathcal{O}(N^k)$, 2-opt and 3-opt are usually preferred to efficiently search for fast improvements of existing tours, although they can get stuck in local optima. Hence, in our work, we use a combination of several local search methods, including local insertion heuristics, random 2-opt, search 2-opt, and search random 3-opt (see below for their definitions), to efficiently find potential 2-opt and 3-opt operations to improve the solution proposed by the RL solver. Instead of running each algorithm once for long enough, we run every local search shortly one by one and repeat this process for multiple times so that the combined local search are able to avoid more local optima. Intuitively, the rationale is that once a certain heuristic gets stuck in a local optimum, another can help get it out by trying a different operation. Next, we explain the above four heuristics, and then illustrate how we leverage them to form our combined local search.

## A.1    RANDOM 2-OPT

For random 2-opt, we randomly try two edges for performing a 2-opt operation and repeat this process for $\alpha \times N^{\beta}$ times, where $\alpha, \beta > 0$ are two hyperparameters controlling the strength of this heuristic as well as its runtime. Theoretically, random 2-opt can potentially cover all 2-opt operations and such a procedure makes random 2-opt much more flexible than trying all $N(N-1)/2$ possible pairs of edges. In this paper, we set $\alpha = 0.5$ and $\beta = 1.5$ for all experiments.

## A.2    LOCAL INSERTION HEURISTIC

For local insertion heuristic, inspired by the insertion heuristic, we iterate through all cities and find the best positions to insert them from their original locations. Let $\sigma$ be the current tour and $\sigma_{t,t'}$ be the tour where we exchange the positions of $\sigma(t)$ with $\sigma(t')$. Namely,

$$\sigma_{t,t'} = \big(\sigma(1), ..., \sigma(t'), \sigma(t), \sigma(t'+1), ..., \sigma(t-1), \sigma(t+1), ..., \sigma(N)\big). \tag{11}$$

for $t' \neq t - 1$ and $\sigma_{t,t-1} = \sigma$. The local insertion heuristic (see Algorithm 1) iterates over every $t \in [N]$. For each $t$, we need to find $t^*$ such that:

$$t^* = \arg\min_{t'} L_{\sigma_{t,t'}}(\boldsymbol{X}), \tag{12}$$

where the definition of $L$ is given in Equation (1). Then we replace $\sigma$ by $\sigma_{t,t^*}$. Theoretically, the local insertion heuristic is a special case of 3-opt where two of the removed edges cover a same node.

---
**Algorithm 1** Local Insertion Heuristic
---
1: **Input**: A matrix of city coordinates $\boldsymbol{X} = (\boldsymbol{x}_i)_{i \in [N]}$, current tour $\sigma$
2: **Output**: An improved tour $\sigma$
3: **for** $t = 1$ **to** $N$ **do**
4: $\quad t^* = \arg\min_{t'} L_{\sigma_{t,t'}}(\boldsymbol{X})$
5: $\quad \sigma \leftarrow \sigma_{t,t^*}$
6: **end for**
7: **return** $\sigma$
---

## A.3 SEARCH 2-OPT

For search 2-opt, we first iterate through all edges and for each first edge, we search for the best second edge to do the 2-opt. Let $\sigma$ be the current tour and $\sigma_{(t,t')}$ be the tour where we reverse the cities between $\sigma(t)$ and $\sigma(t')$. Namely,

$$\sigma_{(t,t')} = \big(\sigma(1), ..., \sigma(t-1), \sigma(t'), \sigma(t'-1), ..., \sigma(t+1), \sigma(t), \sigma(t'+1), ..., \sigma(N)\big). \quad (13)$$

where $t < t'$, and $\sigma_{(t,t)} = \sigma$. Search 2-opt (see Algorithm 2) iterates over every $t \in [N]$. For each $t$, we find $t^*$ such that:

$$t^* = \arg\min_{t' \geq t} L_{\sigma_{(t,t')}}(\boldsymbol{X}) \quad (14)$$

Then we replace $\sigma$ by $\sigma_{(t,t^*)}$. Search 2-opt potentially covers all possible 2-opt operations.

---
**Algorithm 2** Search 2-opt
---
1: **Input**: A matrix of city coordinates $\boldsymbol{X} = (\boldsymbol{x}_i)_{i \in [N]}$, current tour $\sigma$
2: **Output**: An improved tour $\sigma$
3: **for** $t = 1$ **to** $N$ **do**
4: $\quad t^* = \arg\min_{t' \geq t} L_{\sigma(t,t')}(\boldsymbol{X})$
5: $\quad \sigma \leftarrow \sigma_{(t,t^*)}$
6: **end for**
7: **return** $\sigma$
---

## A.4 SEARCH RANDOM 3-OPT

For search random 3-opt, the algorithm first randomly picks two edges. Then for these two randomly-picked edges, similar to search 2-opt, we search to find the best third edge to apply 3-opt. Let $\sigma$ be the current tour, edges from $\sigma(t_1)$ to $\sigma(t_1 + 1)$ and $\sigma(t_2)$ to $\sigma(t_2 + 1)$ be the two randomly picked edges and $\sigma^*_{(t_1,t_2,t_3)}$ be the optimal tour that differs from $\sigma$ only in edges from $\sigma(t_1)$ to $\sigma(t_1+1)$, $\sigma(t_2)$ to $\sigma(t_2+1)$ and $\sigma(t_3)$ to $\sigma(t_3+1)$. For randomly picked $t_1$ and $t_2$, search random 3-opt (see Algorithm 3) iterates over every $t_3 \in [N]$ and find $t_3^*$ such that:

$$t_3^* = \arg\min_{t_3} L_{\sigma^*_{(t_1,t_2,t_3)}}(\boldsymbol{X}) \quad (15)$$

We repeat this process for $\alpha \times N^\beta$ times, where $\alpha, \beta > 0$ are the same two hyperparameters as in random 2-opt. We set $\alpha = 0.5$, $\beta = 1.5$ for all experiments. Search random 3-opt potentially covers all possible 3-opt operation.

## A.5 COMBINED LOCAL SEARCH

With these 4 different local search heuristics, our combined local search applies them one by one sequentially and repeat this for $I$ times (see Algorithm 4), where $I$ is a hyperparameter, which we set to $I = 10$ for all experiments in this paper.

---

**Algorithm 3** Search random 3-opt

---
1: **Input**: A matrix of city coordinates $\boldsymbol{X} = (\boldsymbol{x}_i)_{i \in [N]}$, current tour $\sigma$, hyperparameters $\alpha$ and $\beta$
2: **Output**: An improved tour $\sigma$
3: **for** iter= 1 **to** $\alpha N^\beta$ **do**
4:     Randomly pick $t_1, t_2 \in [N]$ such that $t_1 \neq t_2$
5:     $t_3^* = \arg \min_{t_3} L_{\sigma_{(t_1, t_2, t_3)}^*}(\boldsymbol{X})$
6:     $\sigma \leftarrow \sigma_{(t_1, t_2, t_3^*)}^*$
7: **end for**
8: **return** $\sigma$

---

---

**Algorithm 4** Combined Local Search Algorithm

---
1: **Input**: A matrix of city coordinates $\boldsymbol{X} = (x_i)_{i \in [N]}$, current tour $\sigma$, hyperparameters $\alpha$, $\beta$ and $I$ for local search.
2: **Output**: An improved tour $\sigma$
3: **for** $t = 1$ **to** $I$ **do**
4:     $\sigma \leftarrow$ apply local insertion heuristic$(\boldsymbol{X}, \sigma)$
5:     $\sigma \leftarrow$ apply random 2-opt on $\sigma$ for $\alpha N^\beta$ times
6:     $\sigma \leftarrow$ apply search 2-opt on $\sigma$
7:     $\sigma \leftarrow$ apply search random 3-opt on $\sigma$ for $\alpha N^\beta$ times
8: **end for**
9: **return** $\sigma$

---

# B    POLICY GRADIENT OF EMAGIC

As promised in Section 5 - Smoothed Policy Gradient, we elaborate on the detailed mathematical derivation for Equation (8).

The objective function $J(\boldsymbol{\theta})$ in Equation (3) can be approximated with the empirical mean of the total rewards using $B$ trajectories sampled with policy $\pi_{\boldsymbol{\theta}}$:

$$J(\boldsymbol{\theta}) = -\mathbb{E}[L_\sigma(\boldsymbol{X})] \approx -\hat{\mathbb{E}}_B \left[ \sum_{t=1}^N r_t^{(b)} \right] = -\frac{1}{|B|} \sum_{b=1}^{|B|} \sum_{t=1}^N r_t^{(b)}, \tag{16}$$

where $\hat{\mathbb{E}}_B$ represents the empirical mean operator, $L_\sigma(\boldsymbol{X})$ is the tour length of $\sigma$ output by the RL policy and $r_t^{(b)}$ is the $t$-th reward of the $b$-th trajectory. The policy gradient to optimize $J(\boldsymbol{\theta})$ can be estimated by:

$$\begin{aligned}
\nabla_{\boldsymbol{\theta}} J(\boldsymbol{\theta}) &= -\mathbb{E}_\tau \left[ \left( \sum_{t=1}^N \nabla_{\boldsymbol{\theta}} \log \pi_{\boldsymbol{\theta}}(a_t|\boldsymbol{s}_t) \right) \left( \sum_{t=1}^N r_t \right) \right] \\
&\approx -\hat{\mathbb{E}}_B \left[ \left( \sum_{t=1}^N \nabla_{\boldsymbol{\theta}} \log \pi_{\boldsymbol{\theta}}(a_t^{(b)}|\boldsymbol{s}_t^{(b)}) \right) \left( \sum_{t=1}^N r_t^{(b)} \right) \right]
\end{aligned} \tag{17}$$

However, Recall $J(\boldsymbol{\theta})$ is the standard objective used in most deep RL methods applied to TSP. Instead, we optimize $J^+(\boldsymbol{\theta}) = -\mathbb{E}[L_{\sigma_+}(\boldsymbol{X})]$ where $L_{\sigma_+}(\boldsymbol{X})$ is the tour length of $\sigma$ after applying local search. This helps integrate better RL and local search by smoothing the value landscape and training an RL agent to output a tour that can be improved by local search. This new objective function can be rewritten:

$$\begin{aligned}
J^+(\boldsymbol{\theta}) &= -\mathbb{E}_{\sigma \sim \pi_{\boldsymbol{\theta}}, \sigma_+ \sim \rho(\sigma)}[L_{\sigma_+}(\boldsymbol{X})] \\
&= -\mathbb{E}_{\sigma \sim \pi_{\boldsymbol{\theta}}}[\mathbb{E}_{\sigma_+ \sim \rho(\sigma)}[L_{\sigma_+}(\boldsymbol{X})|\sigma]]
\end{aligned} \tag{18}$$

where $\rho(\sigma)$ denotes the distribution over tours induced by the application of the stochastic local search on $\sigma$. Taking the gradient of this new objective:

$$
\begin{aligned}
\nabla_{\boldsymbol{\theta}} J^+(\boldsymbol{\theta}) &= -\mathbb{E}_\tau\left[\left(\sum_{t=1}^N \nabla_{\boldsymbol{\theta}} \log \pi_{\boldsymbol{\theta}}(a_t|\boldsymbol{s}_t)\right)\mathbb{E}_{\sigma_+\sim\rho(\sigma)}[L_{\sigma_+}(\boldsymbol{X})|\sigma]\right] \\
&\approx -\hat{\mathbb{E}}_B\left[\left(\sum_{t=1}^N \nabla_{\boldsymbol{\theta}} \log \pi_{\boldsymbol{\theta}}(a_t^{(b)}|\boldsymbol{s}_t^{(b)})\right)\left(L_{\sigma_+^{(b)}}(\boldsymbol{X}^{(b)})\right)\right] \\
&\approx -\frac{1}{|B|}\sum_{b=1}^{|B|}\left(\sum_{t=2}^N \nabla_{\boldsymbol{\theta}} \log \pi_{\boldsymbol{\theta}}(a_t^{(b)}|\boldsymbol{s}_t^{(b)})\right)\left(L_{\sigma_+^{(b)}}(\boldsymbol{X}^{(b)})-l^{(b)}\right),
\end{aligned}
\tag{19}
$$

where $\tau = (\boldsymbol{s}_1, a_1, \ldots)$ and $\sigma$ is its associated tour. We simply approximate the conditional expectation over $\rho(\sigma)$ by a sample. Therefore, our gradient estimate is an unbiased estimator of the gradient of our new objective $J^+(\boldsymbol{\theta})$.

Using our policy rollout baseline introduces some bias to the estimation of the smoothed policy gradient. However, the variance reduction helps with achieving greater performance and smaller variance, as we observed in our experiments.

## C  OVERALL ALGORITHM

We provide the pseudo-code of our overall training algorithm in Algorithm 5.

---

**Algorithm 5** REINFORCE exploiting stochastic CL, equivariance, and smoothed policy gradient

---

1: **Input**: Total number of epochs $E$, training steps per epoch $T$, batch size $B$, hyperparameters $\alpha$, $\beta$, $\gamma$ and $I$ for local search
2: Initialize $\boldsymbol{\theta}$
3: **for** $e = 1$ **to** $E$ **do**
4:    $N \leftarrow$ Sample from $\boldsymbol{p}^{(e)}$ according to Stochastic CL
5:    **for** $t = 1$ **to** $T$ **do**
6:        $\forall b \in \{1, ..., B\}\, \boldsymbol{X}^{(b)} \leftarrow$ Random TSP instance with $N$ cities
7:        $\forall b \in \{1, ..., B\}\, \sigma^{(b)} \leftarrow$ Apply $\pi_\theta$ on $\boldsymbol{X}^{(b)}$ after all the equivariant preprocessing steps
8:        $\forall b \in \{1, ..., B\}\, \sigma_+^{(b)} \leftarrow$ Apply the combined local search on $\sigma^{(b)}$
9:        Use $\sigma^{(b)}$ and $\sigma_+^{(b)}$ to calculate $\nabla_{\boldsymbol{\theta}} J^+(\boldsymbol{\theta})$ according to Equation (8)
10:       $\theta \leftarrow$ Update in the direction of $\nabla_{\boldsymbol{\theta}} J^+(\boldsymbol{\theta})$
11:    **end for**
12: **end for**

---

## D  SETTINGS AND HYPERPARAMETERS

All our experiments are run on a computer with an Intel(R) Xeon(R) E5-2678 v3 CPU and a NVIDIA 1080Ti GPU. In consistency with previous work, all our randomly generated TSP instances are sampled in $[0, 1]^2$ uniformly. During training, stochastic CL chooses a TSP size in $\mathcal{R} = \{10, 11, \ldots, 50\}$ for each epoch $e$ according to Equations (9) and (10), with $\sigma_N$ set to be 3. We trained for 200 epochs in each experiment, with 1000 batches of 128 random TSP instances in each epoch. We set the learning rate to be $10^{-3}$ and the learning rate decay to be $0.96$ in each experiment. In each experiment, we set $\alpha = 0.5$, $\beta = 1.5$, $\gamma = 0.25$ and $I = 10$ for the parameters of our combined local search; As for random TSP testing and the ablation study, we test on TSP instances with size 20, 50, 100, 200, 500 and 1000 to evaluate the generalization capability of our model; as for realistic TSP, we test on TSP instances with sizes up to 1002 from the TSPLIB library. With respect to our model architecture, our MLP encoder has an input layer with dimension 2, two hidden layers with dimension 128 and 256, respectively, and an output layers with dimension 128; for the GNN encoder, we set $H = 128$ and $n_{GNN} = 3$.

# E    MORE EXPERIMENTS ON EMAGIC

## E.1    MORE VERSIONS OF EMAGIC

As promised in Section 6, we illustrate all versions of eMAGIC in this section:

Table 4: Results of eMAGIC vs baselines, tested on 10,000 instances for TSP 20, 50, and 100.

| Method | Type† | TSP20 | | | TSP50 | | | TSP100 | | |
|---|---|---|---|---|---|---|---|---|---|---|
| | | Len. | Gap | Time | Len. | Gap | Time | Len. | Gap | Time |
| **eMAGIC(G)** | RL(LS) | 3.841 | 0.29% | 2.8s | 5.732 | 0.74% | 16s | 7.923 | 2.09% | 1.4m |
| **eMAGIC(s1)** | RL(LS) | 3.844 | 0.37% | 3.0s | 5.763 | 1.27% | 17s | 7.964 | 2.61% | 1.3m |
| **eMAGIC(s10)** | RL(LS) | 3.831 | 0.03% | 9.0s | 5.728 | 0.67% | 49s | 7.852 | 1.17% | 2.9m |
| **eMAGIC(S)** | RL(LS) | **3.830** | **0.00%** | 38s | **5.691** | **0.01%** | 3.5m | **7.762** | **0.02%** | 14.6m |

G: Greedily generate a solution from the RL policy and improve it by combined local search.
s1: Randomly sample only one solution from the RL policy and improve it by combined local search.
s10: Randomly sample 10 solutions from the RL policy and improve them by combined local search.
Finally, keep the best tour.
S: Randomly sample 100 solutions from the RL policy and improve by combined local search. Finally, keep the best tour.

Table 5: Results of all versions of eMAGIC, tested on 128 instances for TSP 200, 500, and 1000.

| Method | Type† | TSP200 | | | TSP500 | | | TSP1000 | | |
|---|---|---|---|---|---|---|---|---|---|---|
| | | Len. | Gap | Time | Len. | Gap | Time | Len. | Gap | Time |
| **eMAGIC(G)** | RL(LS) | 11.14 | 3.89% | 36s | 17.52 | 5.89% | 2.0m | 24.702 | 6.85% | 4.9m |
| **eMAGIC(s1)** | RL(LS) | 11.14 | 3.89% | 34s | 17.54 | 6.07% | 1.8m | 24.749 | 7.05% | 4.9m |
| **eMAGIC(s10)** | RL(LS) | 10.95 | 2.12% | 1.1m | 17.29 | 4.50% | 4.1m | 24.503 | 5.99% | 11m |
| **eMAGIC(S)** | RL(LS) | **10.77** | **0.50%** | 2.4m | **17.03** | **2.92%** | 9.7m | **24.126** | **4.36%** | 27m |

See footnotes of Table 4.

## E.2    VARIANCE ANALYSIS OF EMAGIC

As promised in Section 6, we provide the variance analysis of eMAGIC in this section. In our experiments, we repeated all our experiments (training + testing) with 3 random seeds. The variances are shown below:

Table 6: Variance Analysis of eMAGIC

| Model | TSP20 | TSP50 | TSP100 | TSP200 | TSP500 | TSP1000 |
|---|---|---|---|---|---|---|
| **eMAGIC(G)** | 0.043 | 0.049 | 0.050 | 0.064 | 0.056 | 0.066 |
| **eMAGIC(s1)** | 0.0413 | 0.0407 | 0.0509 | 0.0510 | 0.0553 | 0.0655 |
| **eMAGIC(s10)** | 0.0446 | 0.0486 | 0.0434 | 0.0679 | 0.0652 | 0.0713 |
| **eMAGIC(S)** | 0.0478 | 0.0582 | 0.0680 | 0.1207 | 0.1288 | 0.1742 |

See footnotes of Table 4.

The variances are quite low, showing that our method gives relatively good and stable results. Note the variances increase with the number of sampling (s1, s10 and S) for larger TSP instances since there is more room for improvement.

## E.3    EXPERIMENTS ON EXTREMELY LARGE TSP INSTANCES

In Table 7, we show the performances of eMAGIC on extremely large TSP instances (i.e., TSP 10,000) and the comparisons with a few methods that are able to generalize to TSP 10,000. For the hyperparameters of the combined local search, we use $I = 2$ and $\beta = 1.4$. We modify the

hyperparameters in this way because we find that for large TSP instances, doing too many iterations of local search is not efficient. The other experimental settings in this test are the same as Section 6. Also, we test Fu et al. (2021)'s method with a limited the time budget, denoted by AGCRN+M$^{lim}$: since we only reproduce their method successfully with CPU, we decrease their hyperparameter $T$ (the MCTS will end no longer than $T$ seconds) from $0.04n$ to $0.04n/1.8 * (28/60) = 0.010n$ (1.8h and 28m are respectively the runtimes of their model and eMAGIC(G)). By this, we expect that their model and eMAGIC(G) will have a similar running time.

Table 7: Results of eMAGIC vs baselines, tested on 16 instances for TSP 10,000.

| Method | Type[†] | TSP 10,000 | | |
|---|---|---|---|---|
| | | Len. | Gap (to LKH3) | Time |
| LKH3[*] | Heuristic | 70.78 | 0.00% | 8.8h |
| AM[*4] | RL(S) | 431.6 | 501% | 13m |
| AM[*4] | RL(G) | 141.7 | 97.4% | 6.0m |
| AM[*4] | RL(BS) | 129.4 | 80.3% | 1.8h |
| AGCRN+M[*2] | SL+M | 74.92 | 4.39% | 1.8h |
| AGCRN+M[lim] | SL+M | 80.11 | 13.2% | - |
| **eMAGIC(G)** | RL(LS) | 79.28 | 10.5% | 28m |
| **eMAGIC(S)** | RL(LS) | 78.79 | 9.76% | 1.0h |

[*] as reported in previous work.
[2] (Fu et al., 2021), [4] (Kool et al., 2019)
[lim] with a time budget
[†] H - Heuristic; SL - Supervised Learning; RL - Reinforcement Learning; G - Greedy; S - Sampling; M - Monte-Carlo Tree Search; LS - Combined Local Search; BS - Beam Search.

We can observe that our models outperform AM (Kool et al., 2019) to a large extent. Comparing to AGCRN+M (Fu et al., 2021) and LKH3 (Helsgaun, 2017), our methods run much faster without a big gap of performance. Plus, if we give AGCRN+M (Fu et al., 2021) a time budget comparable to our method (i.e., run AGCRN+M and eMAGIC(G) for the same amount of time), we can see that our algorithm outperforms AGCRN+M. Note that MCTS in AGCRN+M is written in C++. We could reduce our runtime if we had also written our code in C++ instead of Python. Therefore, our method offers a better trade-off in terms of performance vs runtime: it can generate relatively good results in much less time.

## F   EVALUATION OF EQUIVARIANT PREPROCESSING METHODS

### F.1   COMPARISONS BETWEEN DIFFERENT SYMMETRIES USED DURING PREPROCESSING

We present the comparison results of applying rotation, translation and reflection. The comparisons are done by testing on random TSP instances followed the same setting in the experiment section. As in table 8, the rotation has the best performance on small and large TSP instances. By this, we choose rotation for our final algorithm.

Table 8: Comparisons between rotation, translation and reflection

| TSP Size | Rotation | | Reflection | | Translation | |
|---|---|---|---|---|---|---|
| | Len. | Gap | Len. | Gap | Len. | Gap |
| TSP 20 | **3.844** | **0.37%** | 3.850 | 0.51% | 3.848 | 0.47% |
| TSP 50 | **5.763** | **1.27%** | 5.786 | 1.67% | 5.807 | 2.05% |
| TSP 100 | **7.964** | **2.61%** | 8.050 | 3.72% | 8.020 | 3.33% |
| TSP 200 | **11.14** | **3.89%** | 11.22 | 4.71% | 11.19 | 4.39% |
| TSP 500 | **17.54** | **6.01%** | 17.65 | 6.69% | 17.68 | 6.85% |
| TSP 1000 | **24.75** | **7.05%** | 24.81 | 7.33% | 24.86 | 7.55% |

### F.2 COMPARISONS BETWEEN ONE PREPROCESSING APPLICATION AND ITERATION PREPROCESSING APPLICATIONS

We present the comparison results of one preprocessing application and iteration preprocessing applications. The comparisons are done by testing on random TSP instances followed the same setting in the experiment section. As in table 9, the iteration preprocessing applications has better performance on small and large TSP instances. By this, we choose iteration preprocessing applications for our final algorithm.

Table 9: Comparisons between iteration preprocessing and one preprocessing

| TSP Size | Iteration[†] | | One[♯] | |
|---|---|---|---|---|
| | Len. | Gap | Len. | Gap |
| TSP 20 | 3.844 | 0.37% | **3.873** | **1.12%** |
| TSP 50 | **5.763** | **1.27%** | 5.871 | 3.17% |
| TSP 100 | **7.964** | **2.61%** | 8.102 | 4.40% |
| TSP 200 | **11.136** | **3.89%** | 11.347 | 5.85% |
| TSP 500 | **17.541** | **6.01%** | 17.742 | 7.23% |
| TSP 1000 | **24.749** | **7.05%** | 24.978 | 8.05% |

[†] Iteration preprocessing application.
[♯] Once preprocessing application.

## G  ABLATION STUDIES

This section illustrates the details of Section 6 - Ablation Study.

### G.1  ABLATION STUDY ON KEY TRAINING TECHNIQUES

In this section, as promised in Section 6 - Ablation Study, we will illustrate the details of the ablation study regarding some key training techniques of eMAGIC. For equivariance, we remove all the equivariant procedures during training/testing (e.g., deleting visited cities, preprocessing, and using relative positions). For the policy rollout baseline, we replace it with the self-critic baseline, which is a greedy baseline implemented in Ma et al. (2019). For combined local search, we only apply it during testing to check if it can help improve the training process. For RL, we directly apply combined local search without performing any learning. Table 10 demonstrates that each technique plays a role in our model.

Table 10: Ablation study on equivariance, policy rollout baseline, combined local search, and RL, tested on 10,000 instances for TSP 20, 50 and 100, and 128 instances for TSP 200, 500 and 1000.

| TSP Size | **Full** | | w/o Equiv.[♯] | | w/o BL[♯] | | w/o LS[♯] | | w/o RL | |
|---|---|---|---|---|---|---|---|---|---|---|
| | Len. | Gap | Len. | Gap | Len. | Gap | Len. | Gap | Len. | Gap |
| TSP20 | **3.844** | **0.37%** | 3.857 | 0.69% | 3.875 | 1.17% | 3.874 | 1.15% | 3.879 | 1.27% |
| TSP50 | **5.763** | **1.27%** | 5.808 | 2.07% | 5.859 | 2.96% | 5.837 | 2.58% | 5.901 | 3.70% |
| TSP100 | **7.964** | **2.61%** | 8.086 | 4.19% | 8.124 | 4.68% | 8.100 | 4.37% | 8.178 | 5.38% |
| TSP200 | **11.14** | **3.89%** | 11.27 | 5.16% | 11.33 | 5.71% | 11.32 | 5.64% | 11.43 | 6.67% |
| TSP500 | **17.54** | **6.01%** | 17.72 | 7.10% | 17.74 | 7.22% | 17.82 | 7.71% | 17.86 | 7.96% |
| TSP1000 | **24.75** | **7.05%** | 24.94 | 7.86% | 24.99 | 8.10% | 25.12 | 8.67% | 25.15 | 8.80% |

[♯] Equiv. - equivariance; BL - Baseline; LS - Combined Local Search.

### G.2  ABLATION STUDY ON OTHER TRAINING TECHNIQUES

In this section, we first perform an ablation study of stochastic CL, meaning we fixed our TSP size to be 50 during training. Moreover, we perform an ablation study on each component of our equivalent model, which includes deleting the visited cities during training, equivariant preprocessing

operation, and using relative positions during training, as mentioned in Section 6 - Ablation Study. For the ablation study of these three components, we remove them during training and testing. Table 11 illustrates the results of our ablation study (including the full version for comparison), which demonstrate that each component is effective in our model.

Table 11: Ablation study on stochastic CL, Deleting, Preprocessing and Relative Position, tested on 10,000 instances for TSP 20, 50 and 100, and 128 instances for TSP 200, 500 and 1000.

| TSP Size | Full | | w/o CL | | w/o Delete | | w/o Pre[†] | | w/o RP[†] | |
|---|---|---|---|---|---|---|---|---|---|---|
| | Len. | Gap | Len. | Gap | Len. | Gap | Len. | Gap | Len. | Gap |
| TSP20 | **3.844** | **0.37%** | 3.861 | 0.79% | 3.855 | 0.66% | 3.855 | 0.64% | 3.852 | 0.57% |
| TSP50 | **5.763** | **1.27%** | 5.821 | 2.30% | 5.824 | 2.34% | 5.811 | 2.11% | 5.823 | 2.33% |
| TSP100 | **7.964** | **2.61%** | 8.062 | 3.88% | 8.064 | 3.90% | 8.084 | 4.16% | 8.093 | 4.28% |
| TSP200 | **11.14** | **3.89%** | 11.29 | 5.29% | 11.28 | 5.18% | 11.27 | 5.17% | 11.30 | 5.44% |
| TSP500 | **17.54** | **6.01%** | 17.69 | 6.93% | 17.67 | 6.80% | 17.68 | 6.85% | 17.72 | 7.07% |
| TSP1000 | **24.75** | **7.05%** | 24.89 | 7.68% | 24.88 | 7.63% | 24.88 | 7.63% | 24.89 | 7.66% |

[†] Pre - Preprocessing; RP - Relative Position.

## G.3 COMPARING PURE RL ALGORITHM WITH FULL ALGORITHM

In Section 6, when doing the ablation study for the combined local search, we do not apply it during training, but we still apply it during testing to check if it can help improve the training process. Here, we compare the pure RL algorithm (full algorithm without local search both in training and testing) and the full algorithm to demonstrate the strength of Combined Local Search:

Table 12: Comparisons between pure RL algorithm and the full algorithm

| TSP Size | Full | | Pure RL | |
|---|---|---|---|---|
| | Len. | Gap | Len. | Gap |
| TSP 20 | **3.844** | **0.37%** | 3.874 | 1.14% |
| TSP 50 | **5.763** | **1.27%** | 6.172 | 8.46% |
| TSP 100 | **7.964** | **2.61%** | 8.688 | 12.0% |
| TSP 200 | **11.136** | **3.89%** | 12.19 | 13.7% |
| TSP 500 | **17.541** | **6.01%** | 19.31 | 16.7% |
| TSP 1000 | **24.749** | **7.05%** | 27.10 | 17.2% |

# H   GNN ENCODER DETAILS

Figure 2 is a detailed illustration of the GNN architecture introduced in Section 4.2. Please notice that Figure 2 could be regarded as a zoom-in version of the GNN part in Figure 1. The aggregation function used in GNN is represented by a neutral network followed by a ReLU function on each entry of the output.

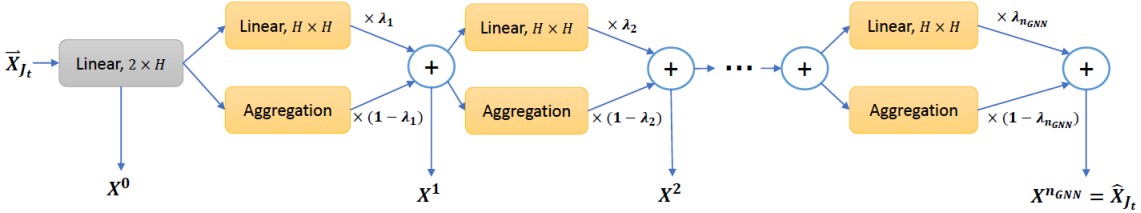

Figure 2: Detailed Architecture of GNN

# I   REALIST TSP INSTANCES IN TSPLIB

As promised in Section 6, Tables 14 to 16 list the performances of various learning-based TSP solvers and heuristic approaches upon realistic TSP instances in TSPLIB. The bold numbers show the best performance among all the approaches. We can observe that most bold numbers are provided by eMAGIC, meaning our approach provides excellent results for TSPLIB. In addition, we also provide our results for the larger instances (with size up to 1002) from TSPLIB in Table 13. Table 13 only contains the experiments of our model since no other model can generalize to large size TSP instances with adequate performance.

Table 13: Comparison of Performance on large size TSP instances in TSPLIB

| Problems | OPT | eMAGIC(S) | | OR-Tools | | Furthest Insertion | |
|---|---|---|---|---|---|---|---|
| | | Length | Gap | Length | Gap | Length | Gap |
| rd400 | 15,281 | **15,707** | **2.79%** | 15,821 | 3.53% | 16,851 | 10.00% |
| f1417 | 11,861 | **11,961** | **0.84%** | 11,996 | 1.14% | 12,845 | 8.23% |
| pr439 | 107,217 | **109,610** | **2.23%** | 117,171 | 9.28% | 121,341 | 12.89% |
| pcb442 | 50,778 | **51,868** | **2.15%** | 52,508 | 3.41% | 57,741 | 13.42% |
| d493 | 35,002 | **36,184** | **3.38%** | 36,599 | 4.56% | 38,869 | 10.69% |
| u574 | 36,905 | 38,486 | 4.28% | **38,467** | **4.23%** | 40,570 | 9.562% |
| rat575 | 6,773 | 7,105 | 4.90% | **6,851** | **1.15%** | 7,632 | 12,09% |
| p654 | 34,643 | **35,001** | **1.03%** | 35,199 | 1.60% | 35,706 | 3.04% |
| d657 | 48,912 | 50,826 | 3.91% | **50,585** | **3.42%** | 53,796 | 9.61% |
| u724 | 41,910 | 43,853 | 4.64% | **43,585** | **4.00%** | 46,367 | 10.16% |
| rat783 | 8,806 | 9,324 | 5.88% | **8,974** | **1.91%** | 9,904 | 11.78% |
| pr1002 | 259,045 | 271,370 | 4.76% | **271,095** | **4.65%** | 285,797 | 9.86% |

Table 14: Comparison of Performances on TSPLIB - Part I

| Problems | OPT | eMAGIC(S) | | Wu et al.[*1] | | S2V-DQN[*2] | |
|---|---|---|---|---|---|---|---|
| | | Len. | Gap | Len. | Gap | Len. | Gap |
| ei151 | 426 | **429** | **0.70%** | 438 | 2.82% | 439 | 3.05% |
| ber1in52 | 7,542 | 7,544 | 0.03% | 8,020 | 6.34% | **7,542** | **0.00%** |
| st70 | 675 | **677** | **0.30%** | 706 | 4.59% | 696 | 3.11% |
| ei176 | 538 | **546** | **1.49%** | 575 | 6.88% | 564 | 4.83% |
| pr76 | 108,159 | **108,159** | **0.00%** | 109,668 | 1.40% | 108,446 | 0.27% |
| rat99 | 1,211 | **1,223** | **0.99%** | 1,419 | 17.18% | 1,280 | 5.70% |
| kroA100 | 21,282 | **21,285** | **0.01%** | 25,196 | 18.39% | 21,897 | 2.89% |
| kroB100 | 22,141 | **22,141** | **0.00%** | 26,563 | 19.97% | 22,692 | 2.49% |
| kroC100 | 20,749 | **20,749** | **0.00%** | 25,343 | 22.14% | 21,074 | 1.57% |
| kroD100 | 21,294 | **21,361** | **0.31%** | 24,771 | 16.33% | 22,102 | 3.79% |
| kroE100 | 22,068 | **22,068** | **0.00%** | 26,903 | 21.91% | 22,913 | 3.83% |
| rd100 | 7,910 | 7916 | 0.08% | **7,915** | **0.06%** | 8,159 | 3.15% |
| ei1101 | 629 | 636 | 1.11% | 658 | 4.61% | 659 | 4.77% |
| 1in105 | 14,379 | **14,379** | **0.00%** | 18,194 | 26.53% | 15,023 | 4.48% |
| pr107 | 44,303 | **44,346** | **0.10%** | 53,056 | 19.76% | 45,113 | 1.83% |
| pr124 | 59,030 | **59,075** | **0.08%** | 66,010 | 11.82% | 61,623 | 4.39% |
| bier127 | 118,282 | **119,151** | **0.73%** | 142,707 | 20.65% | 121,576 | 2.78% |
| ch130 | 6,110 | **6,162** | **0.85%** | 7,120 | 16.53% | 6,270 | 2.62% |
| pr136 | 96,772 | **97,264** | **0.51%** | 105,618 | 9.14% | 99,474 | 2.79% |
| pr144 | 58,537 | **58,537** | **0.00%** | 71,006 | 21.30% | 59,436 | 1.54% |
| ch150 | 6,528 | **6,592** | **0.98%** | 7,916 | 21.26% | 6,985 | 7.00% |
| kroA150 | 26,524 | **26,727** | **0.77%** | 31,244 | 17.80% | 27,888 | 5.14% |
| kroB150 | 26,130 | **26,282** | **0.58%** | 31,407 | 20.20% | 27,209 | 4.13% |
| pr152 | 73,682 | **73,682** | **0.00%** | 85,616 | 16.20% | 75,283 | 2.17% |
| u159 | 42,080 | **42,080** | **0.00%** | 51,327 | 21.97% | 45,433 | 7.97% |
| rat195 | 2,323 | **2,377** | **2.32%** | 2,913 | 25.40% | 2,581 | 11.11% |
| d198 | 15,780 | **15,874** | **0.60%** | 17,962 | 13.83% | 16,453 | 4.26% |
| kroA200 | 29,368 | **29,840** | **1.61%** | 35,958 | 22.44% | 30,965 | 5.44% |
| kroB200 | 29,437 | **29,743** | **1.04%** | 36,412 | 23.69% | 31,692 | 7.66% |
| ts225 | 126,643 | **126,939** | **0.23%** | 158,748 | 25.35% | 136,302 | 7.63% |
| tsp225 | 3,916 | **3,981** | **1.66%** | 4,701 | 20.05% | 4,154 | 6.08% |
| pr226 | 80,369 | **80,436** | **0.08%** | 97,348 | 21.13% | 81,873 | 1.87% |
| gi1262 | 2,378 | **2,417** | **1.64%** | 2,963 | 24.60% | 2,537 | 6.69% |
| pr264 | 49,135 | **49,908** | **1.57%** | 65,946 | 34.21% | 52,364 | 6.57% |
| a280 | 2,579 | **2,635** | **2.17%** | 2,989 | 15.90% | 2,867 | 11.17% |
| pr299 | 48,191 | **48,905** | **1.48%** | 59,786 | 24.06% | 51,895 | 7.69% |
| lin318 | 42,029 | **42,948** | **2.19%** | – | – | 45,375 | 7.96% |

[*] as reported in previous work.
[1] (Wu et al., 2021a), [2] (Khalil et al., 2017)

Table 15: Comparison of Performances on TSPLIB - Part II

| Problems | OPT | L2OPT[*1] | | AM[*2] | | Furthest[3] | |
|---|---|---|---|---|---|---|---|
| | | Len. | Gap | Len. | Gap | Len. | Gap |
| ei151 | 426 | 427 | 0.23% | 435 | 2.11% | 467 | 9.62% |
| ber1in52 | 7,542 | 7,974 | 5.73% | 7,668 | 1.67% | 8,307 | 10.14% |
| st70 | 675 | 680 | 0.74% | 690 | 2.22% | 712 | 5.48% |
| ei176 | 538 | 552 | 2.60% | 563 | 4.64% | 583 | 8.36% |
| pr76 | 108,159 | 111,085 | 2.71% | 111,250 | 2.85% | 119,692 | 10.66% |
| rat99 | 1,211 | 1,388 | 14.62% | 1,319 | 8.91% | 1,314 | 8.51% |
| kroA100 | 21,282 | 23,751 | 11.60% | 38,200 | 79.49% | 23,356 | 9.75% |
| kroB100 | 22,141 | 23,790 | 7.45% | 35,511 | 60.38% | 23,222 | 4.88% |
| kroC100 | 20,749 | 22,672 | 9.27% | 30,642 | 47.67% | 21,699 | 4.58% |
| kroD100 | 21,294 | 23,334 | 9.58% | 32,211 | 51.60% | 22,034 | 3.48% |
| kroE100 | 22,068 | 23,253 | 5.37% | 27,164 | 23.09% | 23,516 | 6.56% |
| rd100 | 7,910 | 7,944 | 0.43% | 8,152 | 3.05% | 8,944 | 13.07% |
| ei1101 | 629 | **635** | **0.95 %** | 667 | 6.04% | 673 | 7.00% |
| lin105 | 14,379 | 16,156 | 12.36% | 51,325 | 256.94% | 15,193 | 5.66% |
| pr107 | 44,303 | 54,378 | 22.74% | 205,519 | 363.89% | 45,905 | 3.62% |
| pr124 | 59,030 | 59,516 | 0.82% | 167,494 | 183.74% | 65,945 | 11.71% |
| bier127 | 118,282 | 121,122 | 2.40% | 207,600 | 75.51% | 129,495 | 9.48% |
| ch130 | 6,110 | 6,175 | 1.06% | 6,316 | 3.37% | 6,498 | 6.35% |
| pr136 | 96,772 | 98,453 | 1.74% | 102,877 | 6.36% | 105,361 | 8.88% |
| pr144 | 58,537 | 61,207 | 4.56% | 183,583 | 213.61% | 61,974 | 5.87% |
| ch150 | 6,528 | 6,597 | 1.06% | 6,877 | 5.34% | 7,210 | 10.45% |
| kroA150 | 26,524 | 30,078 | 13.40% | 42,335 | 59.61% | 28,658 | 8.05% |
| kroB150 | 26,130 | 28,169 | 7.80% | 35,511 | 60.38% | 27,404 | 4.88% |
| pr152 | 73,682 | 75,301 | 2.20% | 103,110 | 39.93% | 75,396 | 2.33% |
| u159 | 42,080 | 42,716 | 1.51% | 115,372 | 174.17% | 46,789 | 11.19% |
| rat195 | 2,323 | 2,955 | 27.21% | 3,661 | 57.59% | 2,609 | 12.31% |
| d198 | 15,780 | — | — | 68,104 | 331.57% | 16,138 | 2.27% |
| kroA200 | 29,368 | 32,522 | 10.74% | 58,643 | 99.68% | 31,949 | 8.79% |
| kroB200 | 29,437 | — | — | 50,867 | 72.79% | 31,522 | 7.08% |
| ts225 | 126,643 | 127,731 | 0.86% | 141,628 | 11.83% | 140,626 | 11.04% |
| tsp225 | 3,916 | 4,354 | 11.18% | 24,816 | 533.70% | 4,280 | 9.30% |
| pr226 | 80,369 | 91,560 | 13.92% | 101,992 | 26.90% | 84,130 | 4.68% |
| gi1262 | 2,378 | 2,490 | 4.71% | 2,683 | 13.24% | 2,623 | 10.30% |
| pr264 | 49,135 | 59,109 | 20.30% | 338,506 | 588.93% | 54,462 | 10.84% |
| a280 | 2,579 | 2,898 | 12.37% | 11,810 | 357.92% | 3,001 | 16.36% |
| pr299 | 48,191 | 59,422 | 23.31% | 513,673 | 938.83% | 51,903 | 7.70% |
| lin318 | 42,029 | — | — | — | — | 45,918 | 9.25% |

[*] as reported in previous work.
[1] (de O. da Costa et al., 2020), [2] Kool et al. (2019), [3] (Furthest Insertion Heuristic)

Table 16: Comparison of Performances on TSPLIB - Part III

| Problems | OPT | OR-Tools[1] | | GPN[2] | | 2-opt | |
|---|---|---|---|---|---|---|---|
| | | Len. | Gap | Len. | Gap | Len. | Gap |
| ei151 | 426 | 436 | 2.35% | 430 | 0.94% | 446 | 4.69% |
| ber1in52 | 7,542 | 7,945 | 5.34% | 8,820 | 16.95% | 7,788 | 3.26% |
| st70 | 675 | 683 | 1.19% | 734 | 8.74% | 753 | 11.56% |
| ei176 | 538 | 561 | 4.28% | 604 | 12,27% | 591 | 9.85% |
| pr76 | 108,159 | 111,104 | 2.72% | 124,404 | 15.02% | 115,460 | 6.75% |
| rat99 | 1,211 | 1,232 | 1.73% | 1,856 | 53.26% | 1,390 | 14.78% |
| kroA100 | 21,282 | 21,448 | 0.78% | 29,676 | 39.44% | 22,876 | 7.49% |
| kroB100 | 22,141 | 23,006 | 3.91% | 31,396 | 41.80% | 23,496 | 6.12% |
| kroC100 | 20,749 | 21,583 | 4.02% | 29,638 | 42.84% | 23,445 | 12.99% |
| kroD100 | 21,294 | 21,636 | 1.61% | 31,343 | 47.19% | 23,967 | 12.55% |
| kroE100 | 22,068 | 22,598 | 2.40% | 33,666 | 52.56% | 22,800 | 3.32% |
| rd100 | 7,910 | 664 | 5.56% | 772 | 22.73% | 702 | 11.61% |
| 1in105 | 14,379 | 14,824 | 3.09% | 24,271 | 68.79% | 15,536 | 8.05% |
| pr107 | 44,303 | 46,072 | 3.99% | 80,744 | 82.25% | 47,058 | 6.22% |
| pr124 | 59,030 | 62,519 | 5.91% | 103,785 | 75.82% | 64,765 | 9.72% |
| bier127 | 118,282 | 122,733 | 3.76% | 190,187 | 60.79% | 128,103 | 8.30% |
| ch130 | 6,110 | 6,284 | 2.85% | 8,785 | 43.78% | 6,470 | 5.89% |
| pr136 | 96,772 | 102,213 | 5.62% | 156,543 | 61.76% | 110,531 | 14.22% |
| pr144 | 58,537 | 59,286 | 1.28% | 116,692 | 99.35% | 60,321 | 3.05% |
| ch150 | 6,528 | 6,729 | 3.08% | 9,973 | 52.77% | 7,232 | 10.78% |
| kroA150 | 26,524 | 27,592 | 4.03% | 47,457 | 78.92% | 29,666 | 11.85% |
| kroB150 | 26,130 | 27,572 | 5.52% | 43,600 | 66.86% | 29,517 | 12.96% |
| pr152 | 73,682 | 75,834 | 2.92% | 145,698 | 97.74% | 77,206 | 4.78% |
| u159 | 42,080 | 45,778 | 8.79% | 87,468 | 107.86% | 47,664 | 13.27% |
| rat195 | 2,323 | 2,389 | 2.84% | 4,960 | 113.42% | 2,605 | 12.14% |
| d198 | 15,780 | 15,963 | 1.16% | 37,267 | 136.17% | 16,596 | 5.17% |
| kroA200 | 29,368 | 29,741 | 1.27% | 61,493 | 109.93% | 32,760 | 11.55% |
| kroB200 | 29,437 | 30,516 | 3.67% | 64,139 | 117.89% | 33,107 | 12.47% |
| ts225 | 126,643 | 128,564 | 1.52% | 265,886 | 109.93% | 138,101 | 9.05% |
| tsp225 | 3,916 | 4,046 | 3.32% | 9,501 | 142.62% | 4,278 | 9.24% |
| pr226 | 80,369 | 82,968 | 3.23% | 198,299 | 146.74% | 89,262 | 11.07% |
| gi1262 | 2,378 | 2,519 | 5.93% | 4,510 | 89.66% | 2,597 | 9.21% |
| pr264 | 49,135 | 51,954 | 5.74% | 151,429 | 208.19% | 54,547 | 11.01% |
| a280 | 2,579 | 2,713 | 5.20% | 6,247 | 142.23% | 2,914 | 12.99% |
| pr299 | 48,191 | 49,447 | 2.61% | 172,390 | 257.72% | 54,914 | 13.95% |
| lin318 | 42,029 | – | – | 103,643 | 146.60% | 45,263 | 7.69% |

* as reported in previous work.
[1] (Perron & Furnon), [2] (Ma et al., 2019)

