# OpenReview forum: "Generalization in Deep RL for TSP Problems via Equivariance and Local Search"
_ICLR.cc/2022/Conference — ICLR 2022 Submitted_

### Official Review · Reviewer_XHWq · 2021-10-22

**Correctness:** 3
**Technical Novelty And Significance:** 2
**Empirical Novelty And Significance:** 3
**Recommendation:** 6
**Confidence:** 4

**Main Review:**

### Strengths
- The use of equivariance properties in deep learning models for TSP is promising and, to the best of my knowledge, new.
- The proposed method show competitive results, generalising to TSP instances of different sizes, which is an important and useful trait that is relatively rare in deep learning approaches to combinatorial optimisation.
- The paper is well written and easy to follow. The experimental details are well explained in the appendix and the authors promised to release their code upon publication, so I see no reproducibility issues.

### Weaknesses
- The main contribution of the paper seems to be the use equivariance properties in the TSP, while other aspects of the model are mainly effective tweaks of existing ideas. In particular, the idea to interleave reinforcement learning with traditional combinatorial optimisation heuristics is not entirely new. A similar approach has already been explored in [1].
- Most if not all of the design choices in the paper are very much tailored for the TSP, so I am not confident the approach is *naturally* extendable to other combinatorial optimisation problems as the authors claim. Such an extension would require a whole other set of equivariance properties, local search heuristics, feature engineering tricks and neural architectures, so it is not even clear what aspects of the method the authors deem transferable to other problems.

### Minor issues
- If I understood it correctly the ablation with respect to the combined local search only considers its influence during training time, and all results include a local search step. It would be interesting to see the gap between the quality of the tour output by the deep learning model and the one refined by local search heuristics.
- In page 3, it reads "A state $s_N$ contains the coordinates of the last unvisited city and those of the first and last visited cities". Should it not be "unvisited cities" instead of "last unvisited city"?

[1] Cai, Qingpeng, et al. "Reinforcement learning driven heuristic optimization." arXiv preprint arXiv:1906.06639 (2019).

**Summary Of The Paper:**

The paper employs equivariance properties and local search heuristics to derive a deep learning model capable of generalising to TSP instances of different sizes. The model constructs a tour, one city at a time, by learning a distribution (policy) over unvisited cities which is optimised via REINFORCE. The solution is further refined with a combination of different local search heuristics, and policy gradients are computed with respect to the final improved solution to smooth the loss landscape. The equivariance in TSP is mostly exploited during preprocessing steps—where rotation, translation and reflection operations are applied to the coordinates of each city—but is preserved by the model architecture which consists of a graph neural network and attention mechanisms. The model is trained with stochastic curriculum learning only on small TSP instances ($\leq 50$ cities) but is shown to have competitive performance on problems with up to 1000 cities.

**Summary Of The Review:**

The paper proposes an effective deep learning model to tackle generalisation in the TSP. While the ideas of the paper are not all novel and somewhat restricted to the TSP, the authors do a great job at integrating previous work on deep learning for combinatorial optimisation into an effective model capable to generalise across problems of different sizes, which is an important challenge in the field. In particular, the use of equivariance properties and the ablation study in the paper should be quite insightful for future work on deep learning for combinatorial optimisation problems.

---

> ### Author Response · Authors · 2021-11-21
> **Author Response to Reviewer XHWq**
>
> Thank you for your review and for noting the competitiveness of our proposition. We address your criticisms and answer your questions below:
>
> **Response to Weakness 1**
>
> Thank you for pointing out the work of Cai et al. After checking their paper, they indeed also train an RL model to initialize a heuristic optimizer. However, there are several differences with our approach:
> - their RL model learns perturbative heuristics, while ours learns constructive heuristics.
> - they do not consider the generalization issue as we do.
> - they focus on bin packing while we focus on TSP.
> - we provide an intuitive explanation of why this approach can work via the derivation of the smoothed policy gradient.
>
>
> **Response to Weakness 2**
>
> Although we demonstrate our approach on TSP, we believe that our general framework can be applied to other combinatorial optimization methods. In particular, for routing problems (e.g., TSP/VRP and variants), our equivariant preprocessing could still be applied and our architecture can be adapted in a similar way as in the attention model [1]. For the local search component, any combination of stochastic heuristics designed for the particular combinatorial optimization problem could be used.
>
> **Response to Minor Issue 1**
>
> We did the extra ablation study on removing the local search both in training and testing. And the results are as follows:
>
> | TSP Size     | Full || Pure RL |  |
> | -----     | ----- | ----- | ----- | -----  |
> |      | Len. | Gap | Len. | Gap |
> |TSP20	|3.844	|0.37%	|3.874	|1.14%|
> |TSP50	|5.763	|1.27%	|6.172 |8.46%|
> |TSP100	|7.964	|2.61%	|8.688 |11.95%|
> |TSP200	|11.14	|3.89%	|12.192|13.74%|
> |TSP500	|17.54	|6.01%	|19.307|16.69%|
> |TSP1000	|24.75	|7.05%	|27.104|17.24%|
>
> We added it in the final version of the paper.
>
>
> **Response to Minor Issue 2**
>
> Thank you for pointing this issue out. We corrected it in the final version of the paper!
>
> [1] Wouter Kool, Herke van Hoof, and Max Welling. Attention, learn to solve routing problems!, 2019.

---

> > ### Comment · Reviewer_XHWq · 2021-11-22
> > **Rebuttal**
> >
> > I would like to thank the authors for their answers. The distinction between their work and Cai et al. is more clear now and the extra ablation study does shed some extra light on the importance of local search.

---

### Official Review · Reviewer_13UN · 2021-11-01

**Correctness:** 3
**Technical Novelty And Significance:** 2
**Empirical Novelty And Significance:** 2
**Recommendation:** 6
**Confidence:** 4

**Details Of Ethics Concerns:**

No.

**Main Review:**

Strengths:
1. According to the experiment results, we can see that the proposed algorithm has better generalizability on large-scale TSP and realistic TSP than most learning-based algorithms.

2. Authors propose an equivariant model to handle the symmetry of the combinatorial structures, which has some novelty.

3. Authors provide detailed experiments and evaluations of different methods and problem instances.

Weakness:
1. Compared with the learning-based algorithm [1] in TSP 200, 500, 1000, the proposed algorithm has longer computational time and larger gaps. Meanwhile, compared with the state-of-art OR solver LKH3, the proposed algorithm seems to only have a very marginal advantage: TSP200 (LKH: 2.0m, eMAGIC(S): 2.4m); TSP500 (LKH: 11m, eMACGIC(S): 9.7m); TSP1000 (LKH3: 38m, eMACGIC(S): 27m). I would encourage the authors to show more evidence to prove that your model has some empirical improvement.

2. In the conclusion section, the authors claim that 'We demonstrated its effectiveness both on random and realistic instances, which shows that our model can reach state-of-the-art performance.' However, based on table 1 and table 2, it looks like both LKH3 and [1] have better performance than the proposed model. I would expect the authors to provide some details about this claim.

[1] Zhang-Hua Fu, Kai-Bin Qiu, and Hongyuan Zha. Generalize a small pre-trained model to arbitrarily large TSP instances. In AAAI, 2021.

**Summary Of The Paper:**

This paper introduces a deep RL approach combined with an equivariant model (to handle Euclidean symmetry) and local search heuristics (to improve a tour) to solve traveling salesman problems (TSP), in particular focusing on the generalizability of large-scale instances. The model consists of a graph neural network (GNN), a multi-layer perceptron (MLP), and an attention mechanism. In the training part, this model involves smoothed policy gradient, and stochastic curriculum learning to speed up the training and make the policy more generalizable. The experiments results show that the proposed approach significantly outperforms most learning-based solvers on large-scale randomly generated TSP and realistic TSP.

**Summary Of The Review:**

Technically, the proposed model uses a standard RL framework with GNN and attention as encoder and decoder. The authors introduce an equivariant model to capture the combinatorial structures of the TSP, which has some novelty and improves the overall performance, but the ablation study does not show that this model provides very significant added value.

Empirically, the proposed model can outperform most learning-based algorithms on large-size TSP but still can not beat the state-of-the-art learning-based algorithm [1] both in computational time and objective values. Meanwhile, compared with the state-of-art OR solver LKH3, the proposed algorithm does not have enough advantage in terms of computational time.

Therefore, my decision is weak reject.

[1] Zhang-Hua Fu, Kai-Bin Qiu, and Hongyuan Zha. Generalize a small pre-trained model to arbitrarily large TSP instances. In AAAI, 2021.

---

> ### Author Response · Authors · 2021-11-21
> **Author Response to Reviewer 13UN**
>
> Thank you for your constructive feedback and for noting the generalizability of our method notably using equivariance. We address your criticisms and questions below:
>
> **Response to Weakness 1**
>
> Our main goal was to propose several techniques that can improve the generalization of deep RL-based solvers. We demonstrate our proposition in the class of deep RL methods for learning constructive heuristics for TSP. As we discussed in our paper, Fu et al.'s method does not belong to that class of methods. Their method performs an adaptive search on the instance to be solved in addition to learning on other TSP instances, while deep RL constructive heuristics only learn on other TSP instances without learning on the current instance. A possible avenue to improve further our method would also be adaptive to the instance to be solved (eg by replacing local search, by MCTS), as we mentioned in our conclusion.
> Moreover, [1] may have a potential memory issue. For testing M TSP cases of size n simultaneously, [1] has an $O(M\times n^2)$ space complexity since they need to store the pairwise heatmap, while ours has only $O(M\times n)$. For testing on 16 cases of TSP 10000, [1] needs 13.4 GB space to store the heatmap.
>
> Also, we would like to recall that not achieving state-of-the-art performance should not be ground for rejection, as mentioned in the ICLR's guide for reviewers. We believe that demonstrating that our proposed techniques can boost generalization in constructive RL methods is an interesting achievement. Having said that, we now provide a further comparison with Fu et al.'s method: their performance may not scale well with a limited time budget. When given a similar time budget, our proposition provides better solutions than Fu et al.'s method. See discussion below and Tables 1 and 2 of our updated submission.
>
> Compared to LKH, our method offers a better trade-off in terms of performance vs runtime: it can generate relatively good results in much less time. To show that, we modified the hyperparameters of the combined local search such that $I=2$ and $\beta=1.4$, and tested on 16 TSP10000 instances. We modify the hyperparameters in this way because we find that for large TSP instances, doing too many iterations of local search is not efficient. We compared the results with [1], [2], and LKH3. Also, we tested [1] with a limited time budget denoted by AGCRN+M$^{lim}$: since we only reproduced [1] successfully with CPU, we decreased their hyperparameter $T$ (the MCTS will end no longer than $T$ seconds) from $0.04n$ to $0.04n / 1.8 * (28/60) = 0.010n$ (1.8h and 28m are the runtime of [1] and eMAGIC(G)) to test one case at a time on CPU. By this, we expect that [1] and eMAGIC(G) will have a similar running time. Note that with their original setting $T=0.04n$, we obtain results similar to their reported ones.
> The results on TSP10000 are provided in the following table:
>
> |Method| Type|   |TSP 10000|   |
> |------|-----|---|---------|---|
> |      |     |Len| Gap (to LKH3)| Time|
> |LKH3*   | H   |70.78|   0.00\%|     8.8h |
> |AM* | RL(S) | 431.6 | 501\% | 13m|
> |AM$^{*}$ | RL(G) | 141.7 | 97.4\%| 6.0m|
> |AM$^{*}$ | RL(BS) | 129.4 | 80.3\% | 1.8h|
> |AGCRN+M$^{*}$ | SL+M | 74.92 | 4.4\% | 1.8h|
> |AGCRN+M$^{lim}$ | SL+M | 80.11 | 13.2\% | -|
> |eMAGIC(G) | RL(LS) | 79.28| 10.5\%| 28m|
> |eMAGIC(S) | RL(LS) | 78.79| 9.76\%| 1.0h|
>
> For eMAGIC(G), we greedily pick the next chosen city without any sampling. We can observe that our models outperform [2] to a large extent. Compared to [1] and LKH3, our algorithm runs much faster without a big gap in performance. With a similar runtime, eMAGIC(G)'s performance is better than [1]. Also, Fu et al.'s MCTS is written in C++.  We could have an even shorter runtime if we had also written our code in C++ instead of Python. We added a discussion of these results in the appendix of the new version of the paper.
>
>
> **Response to Weakness 2**
>
> We should have been more precise in our writing. We meant that among the deep RL-based constructive heuristics, our method reaches state-of-the-art performances, as shown in Tables 1-3. We recall that our model is trained on TSP of sizes 10-50, and then can be used to solve much larger random or realistic TSP instances.
>
> [1] Zhang-Hua Fu, Kai-Bin Qiu, and Hongyuan Zha. Generalize a small pre-trained model to arbitrarily large TSP instances. In AAAI, 2021.
>
> [2] Wouter Kool, Herke van Hoof, and Max Welling. Attention, learn to solve routing problems!, 2019.

---

> > ### Comment · Reviewer_13UN · 2021-11-29
> > **Response to authors**
> >
> > Thanks for your response.
> >
> > I agree that the proposed model can boost generalization in constructive RL methods, which is a significant achievement. I would change my score from 5 to 6.

---

### Official Review · Reviewer_eA3s · 2021-11-01

**Correctness:** 3
**Technical Novelty And Significance:** 3
**Empirical Novelty And Significance:** 2
**Recommendation:** 8
**Confidence:** 5

**Main Review:**

**Strengths**
1. The paper is clear and well-written
2. The proposed preprocessing pipeline is well-motivated and extends nicely previous approaches, in particular by reapplying the preprocessing at each step of the solution construction. It could be leveraged by other learning-based method for solving the TSP.
3. The approach has excellent experimental results:
    * Performance close to (reps. better than) reported state-of-the-art learning-based methods on synthetic (resp. realistic) TSP instances with up to 1000 nodes
    * Good generalisation: one model trained with a range of small instance sizes (up to 50 nodes) is able to generalise to unseen and much larger TSP instances (up to 1000).
    * The ablation study is useful to show the impact of the different components of the approach


**Weaknesses**
1. Missing references to related works, which impacts the novelty claims:
    * “To the best of our knowledge, no other work uses curriculum learning for designing an RL-based solver for TSP.” See reference [1] below.
    * The authors claim that (to the best of their knowledge) they are the first to update the encoding of the state to take into account the removal of the visited nodes. This has been explored in (at least in) [2].
2. The proposed approach is presented as a novel approach that exploits “equivariance” but it seems to me that it mostly exploits the properties of the TSP (indeed invariance of the solution to certain transformations of the coordinates) to preprocess the instances. Then the only equivariance that is exploited by the model is with respect to the node ordering, by using a GNN, which is a pretty standard choice of architecture (see e.g. cited [Joshi et al 2019], [Fu et al 2020]).
3. Overall, the proposed approach is quite specific to the TSP (the preprocessing, the local search and the architecture), it is not straightforward how much of it could be exploited for solving other CO problems.
4. Some imprecisions/missing information, see Questions.

[1] Lisicki et al. "Evaluating Curriculum Learning Strategies in Neural Combinatorial Optimization." Workshop on Learning Meets Combinatorial Algorithms @ NeurIPS 2020


[2] Peng et al, A Deep Reinforcement Learning Algorithm Using Dynamic Attention Model for Vehicle Routing Problems. International Symposium on Intelligence Computation and Applications. 2019

**Questions**
1. The form of the policy gradient in equation (8) could be interpreted as the standard policy gradient with as baseline the value of the solution after applying local search. Since the log probabilities correspond to the model’s output solution and not the improved one by local search. Can you comment if this interpretation is correct or not?
2. Given the form of the scores in (9), as normal distributions with mean e the index of the current epoch, when (if) e becomes much bigger than 50 during training, the probability of sampling graphs of size 50 or 10 will be quite close, right? Is it intended? What is the number of epochs range?
3. What is the impact on training time of adding the local search, updating the encodings of all nodes and the preprocessing?
4. Table 1: The results for GCN* are different (a bit worse) from what is reported in the paper [Joshi et al 2019b]. Why is there a difference and why not use the beam-search version of their method the gives better results for running times similar to the proposed eMAGIC?
5. In general, what is the motivation behind the chosen baselines? Versus more recent approaches such as [3]? Why are the baselines not the same in Tables 1-2 and 3? Why not include Concorde and LKH results in Table 3?
6. Table 2: what is the difference between GPN^*5 and GPN^5?
7. Table 3:
    * what are the running times? this is key to be able to compare the different methods.
    * why are the results of Furthest insertion and OR tools not reported for instances with 400-1002 nodes?


[3] Kwon et al, POMO: Policy Optimization with Multiple Optima for Reinforcement Learning, NeurIPS 2020



**Additional feedback**

Minor typos, mathematical inconsistencies and suggestions
* Introduction:  “equivariance for e” etc —> “e for equivariance” (and same for the others) would be a more standard acronym definition
* Sec 3: “this RL problem corresponds to a repeated N-horizon sequential decision-making problem.” —> repeated (N-t)-horizon ?
* Sec 4.1: “and any coordinates x after preprocessing by $\tilde{x}$” —> missing bold
* Sec 4.2
    * Should \tilde{x}_{\sigma(t-1)} be replaced by \tilde{x}_{\sigma(t)} in the definition of the vector $x$ at $\sigma(n)$? (As in the definition for $\sigma(1)$)
    * The MLP encodes the information about visited cities —>  about the first visited city?
    * As a GNN, it is invariant with respect to .. —> equivariant
* Sec 5: It seems the sign of the gradient in equation (8) is inconsistent with what is presented in Appendix B
* Sec 6
    * …not being to solve
    * Tables 1 and 2:
        * the model from [Kool et al 2019] could be called AM instead of GAT for clarity
        * Putting the best optimality gap in bold would be helpful
    * …in the their papers




**Summary Of The Paper:**

The paper proposes a series of techniques and a new model to learn to solve the TSP using RL. First, a number of preprocessing steps are presented in order to transform a TSP instance into a standardized form. Then, a new encoder-decoder architecture is proposed, based on GNN, MLP and attention modules. The authors propose a modification of the policy gradient algorithms for training by replacing the value of the policy-generated solution by its value after applying a local search heuristic, and use the original value as a baseline. Finally they use a curriculum learning approach to increase the difficulty of the instances seen during training, the difficulty here being represented by increasing the size of the instances, within a given interval. Experimental results are presented on standard synthetic and realistic TSP instances of size up to 1000 nodes.


**Summary Of The Review:**

I would vote for reject. My main concern is about the applicability of the proposed contributions beyond the TSP. Clearly the proposed approach is great for the TSP, but it is also somewhat hand-crafted for it (see weaknesses). To me the significance of a TSP-specific learning technique is limited — unless it clearly outperforms specialised TSP solvers such as Concorde or LKH, which is not claimed here. The ultimate goal of learning-based heuristics for combinatorial problems is to replace the handcrafted problem-specific techniques by end-to-end ones. Proving the relevance of the contributions for solving other CO problems would make the paper much stronger.


--------
### Update after rebuttal

I have appreciated the high quality of the authors rebuttal. My concerns and questions were properly addressed, in particular regarding
* the explanation of the proposed smoothed gradient,
* the novelty with respect to previous works,
* and the applicability of the proposed contributions beyond the TSP.

Overall, I think the paper provides a set of well-motivated contributions that lead to an RL-based TSP heuristic with a significant improvement in generalisation performance, compared to current “similar” methods. Some of the proposed techniques could be adapted to other CO problems/ML-based CO heuristics. Therefore I believe this paper can be interesting for the NCO community to participate in addressing the well-recognized problem of generalization of ML-based CO solvers.

I am happy to revise my score from 5 (marginally below acceptance threshold) to 8 (accept good paper).

---

> ### Author Response · Authors · 2021-11-21
> **Author Response to Reviewer eA3s (Part1)**
>
> Thank you for your time and for noting the excellent performance of our proposition. We address your criticisms and answer your questions below:
>
> **Response to Weakness 1**
>
> Thank you for sharing those references. We now discussed them in the final version.
> Compared to Lisicki et al., we propose a stochastic curriculum learning method. Moreover, we deal with much larger TSP instances.
> Our idea of removing visited cities is indeed similar to Peng et al., however, we explore this technique in the context of generalization to large instances. Moreover, our proposition integrates several other novel ideas, e.g., iterative preprocessing or smoothed policy gradient.
>
> **Response to Weakness 2**
>
> Our goal in this paper is to formulate a general framework (equivariant preprocessing+smoothed policy gradient+interleaving policy updates and local search+stochastic curriculum learning) that can train an RL solver on small instances and generalize to solve larger instances. Hence, we identify equivariance as a general property for preprocessing.
> When instantiating our approach to TSP, equivariance happens on several occasions, notably:
> - independence to city permutation, as already noted
> - scaling of city positions requires a corresponding transformation of the rewards
> - removing visited cities requires a corresponding transformation in the output of the RL policy, and another of the returns
>
> **Response to Weakness 3**
>
> Although we demonstrate our approach on TSP, we believe that our general framework can be applied to other combinatorial optimization methods. In particular, for routing problems (e.g., TSP/VRP and variants), our equivariant preprocessing could still be applied and our architecture can be adapted in a similar way as in the attention model [1]. For the local search component, any combination of stochastic heuristics designed for the particular combinatorial optimization problem could be used.
>
> **Response to Question 1**
>
> Your interpretation is correct. We just change the returns, but not the log probabilities. We propose an interpretation of this new objective (ie smoothing of the value landscape) in the paragraph before (8).
>
> **Response to Question 2**
> Good point! As mentioned in Appendix D, we set the number of epochs to 200. Since we focus on generalization, we noticed that training on instances of diverse sizes helps not to overfit to instances of one size.
>
> **Response to Question 3**
> The training time does increase when we add local search, update the encodings of all nodes, or apply to preprocess. We found that local search contributes the most in the increase of the training time, while all the other procedures have little impact. However, training time is not our main concern because our goal is to obtain a fast and generalizable TSP model.
>
> For your information, we provide below the training time increases when adding local search: for an epoch with TSP20 instances, the training time increases from the 40s to 1 minute; for an epoch with TSP40 instances, the training time increases from 2 minutes to 4 minutes (with hyperparameters setting indicated in the paper).
>
> **Response to Question 4**
>
> We used the results reported in Fu et al. We did not focus much on comparing with the different versions of GCN because (1) we wanted to compare mainly with other RL methods similar to ours (RL used as a constructive heuristics) and (2) GCN does not perform very well in terms of generalization as shown in Table 2. However, we can definitely add such comparisons in the final version:
>
> | Model     |       | TSP20 ||  | TSP50 |   | | TSP100||
> | -----     | ----- | ----- | ----- | -----  |  ----- | -----   |-----   |-----   |-----   |
> |      | Len| Gap |Time| Len| Gap |Time |Len| Gap |Time|
> | GCN(BS) |3.835| 0.12% |21m |5.707 |0.29%| 35m|7.876 |1.48% |32m |
> | GCN(BST) |3.831| 0.01%| 22m| 5.692| 0.03%| 38m |7.872 |1.43%| 1.2h|
> | eMAGIC(G) | 3.841| 0.29\%  |2.8s|     5.732  | 0.74\%  | 16s    | 7.923  | 2.09\%   | 1.4m|
> | eMAGIC(S) |3.830| 0.00% |38s |5.691 |0.01%| 3.5m| 7.762 |0.02% |15m|
>
> | Model     |       | TSP200 ||  | TSP500 |   | | TSP1000||
> | -----     | ----- | ----- | ----- | -----  |  ----- | -----   |-----   |-----   |-----   |
> |      | Len| Gap |Time| Len| Gap |Time |Len| Gap |Time|
> | GCN(BS) |16.19|51.02\%| 4.6m|30.37 |83.55\%| 38m|51.26 |122\%| 52m |
> | GCN(BST) |16.21| 51.21\%| 4.0m| 30.43 |83.89\%| 31m |51.10 |121\%| 3.2h|
> | eMAGIC(G) | 11.14  | 3.89\%  | 36s    | 17.52 | 5.89\% | 2.0m   | 24.70| 6.85\%  |4.9m |
> | eMAGIC(S)|  10.77 |f 0.50\% |2.4m |17.03 | 2.92\% |    9.7m  |24.13 | 4.36\%  | 27m |
>
> where BS corresponds to beam search and BST to Beam search and Shortest tour heuristic.
>
> As can be seen, although GCN(BS) performs a little better than our model in Table 1 (small size instances), their runtime is much longer than ours. If we use sampling in our model, our method can beat GCN(BS) both in terms of runtime and performance.

---

> > ### Author Response · Authors · 2021-11-21
> > **Author Response to Reviewer eA3s (Part2)**
> >
> > **Response to Question 5**
> >
> > Thank you for pointing out the POMO model [2]! We added a reference to it in our final version. Note that their work does not focus on generalization and the largest TSP instances their approach is applied on are TSP100.
> >
> > The performances of the baselines mainly come from previously reported results. We believe that they clearly suggest that our RL agent, which is only trained on TSP10-50, is competitive and generalizes to both large TSP instances and realistic instances.
> >
> > For the question why the baselines are not the same in Tables 1-2 and 3, please refer to the response to question 3 of reviewer n7h8.
> >
> > Concorde and LKH generally give optimal solutions to almost all the TSPLib instances, but with a very large runtime. That is why we did not include them.
> >
> > **Response to Question 6**
> >
> > GPN$^*$ represents the results reported by the GPN paper [3] (* means the results come from previous papers), and GPN represents the results that we obtained using their published codes. We did our best to run their algorithm, but could not achieve the good performances reported in their paper. We contacted the authors about this issue, but we didn't obtain any response.
> > We reported our experimental results of GPN to give an idea of its running time.
> >
> > **Response to Question 7**
> >
> > Our goal for testing on TSPLib was to show that our method can generalize from training on small random TSP instances to realistic TSP instances. We believe that in terms of performance, our method is very competitive among learning-based methods. The runtime for our method is below 5 minutes for the larger instances and below 2 minutes for smaller instances.
> >
> > We also run extra experiments for Furthest Insertion and OR-Tools on cases ranging from 400 to 1002 in TSPLib:
> >
> > | Range     | eMAGIC(S) |Furthest Insertion |OR-Tools|
> > | -----     | ----- | ----- |----- |
> > |  400-1002   | 3.40% | 10.11%	| 3.57% |
> >
> > The detailed results are added in the appendix. From the results above, we have a good generalization ability that we perform better on large cases compared to Furthest Insertion and OR-Tools. For other baselines, they have a worse gap when the sizes of problems become bigger. Notice that our average gap for the instances with sizes from 400 to 1002 is less than their average gaps for smaller problems, thus indicating that we can expect to have a smaller average gap when the problem sizes are in the range of 400 to 1002. This suggests further the greater generalization ability of our proposition.
> >
> >
> > **Response to Additional Feedbacks**
> >
> > Thank you for reviewing our paper so carefully! We corrected all the issues in the final version.
> >
> > [1] Wouter Kool, Herke van Hoof, and Max Welling. Attention, learn to solve routing problems!, 2019.
> >
> > [2] Kwon et al, POMO: Policy Optimization with Multiple Optima for Reinforcement Learning, NeurIPS 2020.
> >
> > [3] Qiang Ma, Suwen Ge, Danyang He, Darshan Thaker, and Iddo Drori. Combinatorial optimization
> > by graph pointer networks and hierarchical reinforcement learning. CoRR, 2019.

---

### Official Review · Reviewer_n7h8 · 2021-11-02

**Correctness:** 3
**Technical Novelty And Significance:** 3
**Empirical Novelty And Significance:** 3
**Recommendation:** 5
**Confidence:** 4

**Main Review:**

Originality: Two contributions are original. The first one is algorithmic, i.e., including local search in the training loop of policy gradient. The second is engineering, i.e., applying equivariant preprocessing at each step instead of only once at the beginning. With these two novelties, I think this paper clears the bar for originality.

Quality: Overall this paper is of good quality. I have several questions on the technical details:
1. Could the authors provide a more rigorous derivation for the smoothed policy gradient formula (Equation 8). I did read section B in the Appendix but it jumped from Equation 17, which is the normal policy gradient, to Equation 8 without much explanation. Specifically, is Equation 8 an unbiased gradient estimator for the objective $J^{+}(\theta)$?

A second question on the gradient computation is about the baseline, which is the value of the rollout tour from the policy. Does subtracting this baseline still yield an unbiased gradient estimator?

2. For the empirical studies, can the authors comment on how many random seeds are used to obtain the RL results and what are the variances among different random seed runs?

3. I also noticed that the compared methods are different for random TSP instances and TSPLIB instances. Can the authors provide an explanation?

Clarity: This paper is well-written and easy to understand. Good job!

Significance: Since I have some important questions regarding the technical contributions, I will need to see the authors’ response in order to have a more informed evaluation of the significance of the paper. In its current state, I think the significance is not enough because of some (potential) technical issues and empirical baseline choices.


**Summary Of The Paper:**

This paper proposes a novel combination of policy gradient and local search algorithms for the traveling salesman problems (TSP). The main idea is to apply local search to tours generated by policy rollout and compute policy gradients using the (potentially) improved tour from local search. In addition to the algorithmic contribution, this paper also presents a collection of preprocessing steps to ensure problem instance features are equivalent. Empirical studies are provided on random TSP instances as well as those in TSPLIB. Finally, ablation studies show the importance for each component of the proposed algorithm.

**Summary Of The Review:**

My current recommendation for the paper is marginally below the acceptance threshold. I like the novel ideas in the paper and I admire the thoroughness to include lots of baselines in the random TSP instances comparison. However, there are some unclear points in the algorithmic details. And for the more realistic TSPLIB instances, the number of baselines is much smaller. These two points left me some doubts about the correctness and the significance of the paper.

---

> ### Author Response · Authors · 2021-11-21
> **Author Response to Reviewer n7h8 (Part1)**
>
> Thank you for your review and for noting the quality of our paper. We answer your questions below:
>
> **Response to question 1:**
>
> Recall $J(\boldsymbol\theta) = -\mathbb E[L_\sigma(\boldsymbol X)]$ is the standard objective used in most deep RL methods applied to TSP, where $L_\sigma(\boldsymbol X)$ is the tour length of $\sigma$ output by the RL policy. Instead, we optimize $J^+(\boldsymbol\theta)= -\mathbb E[L_{\sigma_+}(\boldsymbol X)]$ where $L_{\sigma_+}(\boldsymbol X)$ is the tour length of $\sigma$ after applying local search. This helps integrate better RL and local search by smoothing the value landscape and training an RL agent to output a tour that can be improved by local search.
> This new objective function can be rewritten:
> $$
> \begin{aligned}
> J^+(\boldsymbol\theta) &= -\mathbb E_{\sigma \sim \pi_\boldsymbol\theta\\,,\\,\\,\sigma_+\\,\sim\rho(\sigma)}\\,[L_{\sigma_+}(\boldsymbol X)]\\\\
> &= -\mathbb E_{\sigma \sim \pi_\boldsymbol\theta}\\,[\mathbb E_{\sigma_+\\,\sim\rho(\sigma)}\\,[L_{\sigma_+}(\boldsymbol X) | \sigma]]
> \end{aligned}
> $$
> where $\rho(\sigma)$ denotes the distribution over tours induced by the application of the stochastic local search on $\sigma$.
> Taking the gradient of this new objective:
>
> $$
> \begin{aligned}
> \\nabla_{\\boldsymbol{\\theta}} J^+(\\boldsymbol{\\theta}) &= - \\mathbb E_\\tau \left[ \left( \\sum_{t=1}^{N} \\nabla_{\\boldsymbol\\theta}  \\log \\pi_{\\boldsymbol\\theta}(a_t | \boldsymbol{s}_t)\right) \\mathbb E_\{\\sigma_\+\\,\\sim\\rho(\\sigma)\} \left[ L_\{\\sigma_\+\}(\\boldsymbol X)  | \\sigma \right] \right] \\\\
> & \\approx - \hat{\mathbb E}_B\left[\left( \sum_\{t=1\}^N \\nabla_\{\\boldsymbol\\theta\} \log \pi_\{\\boldsymbol\\theta\}(a^{(b)}_t | \\boldsymbol{s}^{(b)}_t)\right)\left(L_\{\\sigma^{(b)}_\+}\\,(\\boldsymbol X^{(b)})\right)\right]
> \end{aligned}
> $$
>
> where $\tau = (\boldsymbol{s}_1, a_1, \ldots)$ and $\sigma$ is its associated tour.
> We simply approximate the conditional expectation over $\rho(\sigma)$ by a sample. Therefore, our gradient estimate is an unbiased estimator of the gradient of our new objective $J^+(\boldsymbol\theta)$.
>
> Using our policy rollout baseline introduces some bias to the estimation of the smoothed policy gradient, however, the variance reduction helps with achieving greater performance, as we observed in our experiments.
>
> Thank you for this question, we added this discussion in the final version, which indeed clarifies the derivation of the smoothed policy gradient and its properties.
>
> **Response to question 2:**
>
> When reporting our results, we followed previous practices [1, 2, 3, 4], which do not provide this information. In our experiments, we repeated all our experiments (training + testing) with 3 random seeds. The variances are shown below:
>
> | Model     | TSP20 | TSP50 | TSP100| TSP200 | TSP500 | TSP1000 |
> | -----     | ----- | ----- | ----- | -----  |  ----- | -----   |
> |eMAGIC(G) |0.043| 	0.049| 	0.050| 	0.064| 	0.056 	|0.066|
> | eMAGIC(s1）    | 0.0413 	| 0.0407 	| 0.0509 	| 0.0510 	| 0.0553 	| 0.0655 |
> | eMAGIC(s10) | 0.0446 	| 0.0486 	| 0.0434 	| 0.0679 	| 0.0652 	| 0.0713 |
> | eMAGIC(S) | 0.0478 	| 0.0582 	| 0.0680 	| 0.1207 	| 0.1288 	| 0.1742 |
>
>
> The variances are quite low, showing that our method gives relatively good and stable results. Note the variances increase with the number of sampling (s1, s10, and S) for larger TSP instances since there is more room for improvement.
> We added this table in the appendix.

---

> > ### Author Response · Authors · 2021-11-21
> > **Author Response to Reviewer n7h8 (Part2)**
> >
> > **Response to question 3:**
> >
> > For fairness, we chose to use the reported results for previous methods (marked with a star in Tables 1, 2, and 3) when we could, since our experimental set-up is the same and their respective authors have normally already tried to achieve the best possible results for their methods.
> > Moreover, we also noticed that reproducing some reported results was sometimes challenging despite our best effort. For instance, the source code of GPN does not exactly correspond to its description in their paper. The source code of Fu et al. uses a very old version of CUDA. Having said that, we reported the results of our run of GPN in Tables 1 and 2 in order to give an idea of its runtime.
> >
> > For TSPLib, most previous work does not evaluate their approaches in those realistic instances. As additional baselines, we ran GPN, furthest insertion, and OR-tools on TSPLib for more comparison, and the results are as followed:
> >
> > | Range     | GPN |
> > | -----     | ----- |
> > |  50-199   | 56.46% |
> > | 200-399   | 147.10%  |
> >
> > | Range     | Furthest Insertion | OR-Tools |
> > | -----     | ----- | ----- |
> > |  400-1002   | 10.11% |3.57%|
> >
> >
> > The detailed results can be found in Tables 14-16 in the appendix. Overall, they show that eMAGIC performs much better than GPN and furthest insertion, and we still beat OR-tools when the size of the problem is even larger. The runtime for our method is below 5 minutes for the larger instances and below 2 minutes for smaller instances.
> >
> > [1] Zhang-Hua Fu, Kai-Bin Qiu, and Hongyuan Zha. Generalize a small pre-trained model to arbitrarily large TSP instances. In AAAI, 2021.
> >
> > [2] Wouter Kool, Herke van Hoof, and Max Welling. Attention, learn to solve routing problems!, 2019.
> >
> > [3] Qiang Ma, Suwen Ge, Danyang He, Darshan Thaker, and Iddo Drori. Combinatorial optimization by graph pointer networks and hierarchical reinforcement learning. CoRR, 2019.
> >
> > [4] Chaitanya K. Joshi, Thomas Laurent, and Xavier Bresson. An efficient graph convolutional network technique for the traveling salesman problem. C

---

### Author Response · Authors · 2021-11-21
**Comment to all reviewers**

We thank all the reviewers for their comments that helped us clarify the presentation of the paper. We now discuss all the missing related work that the reviewers brought to our attention. We would like to restate the objective of our paper which was to demonstrate that generalization in constructive deep RL can be achieved by using a combination of simple techniques. Although some ideas were proposed before and evaluated in different context, our proposed method still includes several novelties in our opinion:
- systematic exploitation of equivariance, notably with repeated preprocessing
- simplified neural network architecture thanks to equivariance
- intuitive motivation for smoothed policy gradient and interleaving policy update and local search
- novel stochastic curriculum learning technique

Based on all the feedback, we have updated our submission. All the changes are written in orange in the new pdf for ease of inspection. We would like to summarize what we have added and modified below:
1. We added a more detailed derivation for the smoothed policy gradient formula in the appendix.
2. We added eMAGIC(G) and changed some notations: eMAGIC(G) greedily chooses the next city from the probability given by the attention mechanism without any sampling; changed eMAGIC to eMAGIC(s1) and eMAGIC(s) to eMAGIC(s10). Since the results for eMAGIC(s1) and eMAGIC(s10) may not be as interesting as eMAGIC(G), we put them in the appendix to save space.
3. We added the variance analysis for eMAGIC(G), eMAGIC(S), eMAGIC(s1) and eMAGIC(s10) in the appendix.
4. We additionally tested GPN [1] on TSPLib, and tested Furthest Insertion and OR-Tools on problems with size range from 400-1002. The results are included in the appendix.
5. We updated the related work section to add some discussion on the missing previous work.
6. We added the explanation of $^*$ in the footnotes of the tables, which we forgot to add in the previous version of the paper. We are sorry for the potential confusion. For fairness, we chose to use the reported results for previous methods (marked with a star in Tables 1, 2, and 3) when we could, since our experimental set-up is the same and their respective authors have normally already tried to achieve the best possible results for their methods.
7. We added GCN(BS) and GCN(BST) in Tables 1 and 2 to have more baselines.
8. We tested eMAGIC(G) and eMAGIC(S) on random TSP10000. We compared our results with LKH3, [2], [3], and [3] with a limited time budget. The details are included in the appendix. In summary, our approach can achieve good results with a much shorter runtime than LKH3 and [3], and we achieve better results compared to [3] with a comparable time budget. Those new experimental results further show that our method offers a better trade-off in terms of performance vs runtime: it can generate relatively good results in much less time.
9. We added one more ablation study in the appendix where our combined local search is removed both in training and testing.
10. We fixed all the typos and minor issues raised by the reviewers.

[1] Qiang Ma, Suwen Ge, Danyang He, Darshan Thaker, and Iddo Drori. Combinatorial optimization by graph pointer networks and hierarchical reinforcement learning. CoRR, 2019.

[2] Wouter Kool, Herke van Hoof, and Max Welling. Attention, learn to solve routing problems!, 2019.

[3] Zhang-Hua Fu, Kai-Bin Qiu, and Hongyuan Zha. Generalize a small pre-trained model to arbitrarily large TSP instances. In AAAI, 2021.

---

### Author Response · Authors · 2021-11-25
**Concerns about whether the revisions addressed the reviewers' questions?**

We sincerely thank all the authors for pointing out a number of good questions for our paper. Our paper becomes more precise and some mistakes are corrected after the revision. We would appreciate reviewers' replies indicating whether our response and clarifications have addressed the issues raised in the reviews, or whether there is anything else we can possibly further clarify.

---

> ### Comment · Reviewer_XHWq · 2021-11-29
> **Overall impression after rebuttal**
>
> I would like to thank the authors for the improvements made in the paper. I believe they have addressed most of the concerns put forward during this reviewing process, but I remain skeptical about the generalisation of the proposed framework to other combinatorial optimisation problems.  While the paper thoroughly studies the TSP, introducing a promising model and achieving results comparable to state-of-the-art machine learning methods (Fu et al. 2021), the equivariant preprocessing steps are highly problem-specific, and their relevance to other problems beyond the TSP is hard to gauge. That remains to me the main weakness of the paper, as new TSP-specific solvers are not really interesting in themselves unless they outperform Concorde, which offers convergence guarantees besides better solutions. The same exclusive focus on the TSP is observed in other machine learning approaches, but then solution quality and generalisation ability similar to eMAGIC are also observed in previous work, notably Fu et al. 2021. All in all, this is an interesting and very well executed paper proposing a new learnable TSP solver, but given the concerns above about its overall relevance to other combinatorial optimisation problems, I am inclined to maintain my original score 6.

---

### Decision · Program_Chairs · 2022-01-20

**Decision:**

Reject

**Comment:**

The paper discusses new RL algorithms for solving large. TSP instances. The algorithm is novel and the problem is important however certain technical questions regarding the soundness of the algorithm were raised. Furthermore, it seems that despite much larger computational time, the algorithm provides only very moderate gains over previous baselines. Finally, it is not clear how the proposed methods (e.g. equivariance) can be applied outside of the TSP problem scope. Thus the concern is the limited impact of the method on the field.

The authors addressed some of the concerns of the reviewers in the rebuttal however it is still not clear:

(1) how the presented mechanism can be applied for other combinatorial problems beyond TSP and therefore how useful it can be for the machine learning community,

(2) how novel the paper is (the use of equivariance is as direct as in the regular graph neural network setup).

Furthermore, the experiments show that the deep learning approach to TSP is still not competitive with standard non-machine baselines. Thus it is not clear whether the proposed algorithm is a right approach to solve this problem, even though it beats other deep learning techniques. The paper is very well written though and the presented method is definitely of value to the research community working on the TSP. Therefore it seems that at this point the paper is more suited for one of the mathematical journals on combinatorics and graph theory.